# The Destructive Fungal Pathogen *Botrytis cinerea*—Insights from Genes Studied with Mutant Analysis

**DOI:** 10.3390/pathogens9110923

**Published:** 2020-11-07

**Authors:** Nicholas Cheung, Lei Tian, Xueru Liu, Xin Li

**Affiliations:** 1Michael Smith Laboratories, University of British Columbia, Vancouver, BC V6T 1Z4, Canada; nicholascl.cheung@alumni.ubc.ca (N.C.); lei.tian@msl.ubc.ca (L.T.); xueru.liu@alumni.ubc.ca (X.L.); 2Department of Botany, University of British Columbia, Vancouver, BC V6T 1Z4, Canada

**Keywords:** airborne fungal pathogen, *Botrytis cinerea*, fungal pathogenesis, sclerotial development, fungal growth, conidiation, melanization

## Abstract

*Botrytis cinerea* is one of the most destructive fungal pathogens affecting numerous plant hosts, including many important crop species. As a molecularly under-studied organism, its genome was only sequenced at the beginning of this century and it was recently updated with improved gene annotation and completeness. In this review, we summarize key molecular studies on *B. cinerea* developmental and pathogenesis processes, specifically on genes studied comprehensively with mutant analysis. Analyses of these studies have unveiled key genes in the biological processes of this pathogen, including hyphal growth, sclerotial formation, conidiation, pathogenicity and melanization. In addition, our synthesis has uncovered gaps in the present knowledge regarding development and virulence mechanisms. We hope this review will serve to enhance the knowledge of the biological mechanisms behind this notorious fungal pathogen.

## 1. Introduction

Ascomycete *Botrytis cinerea* is a fungal pathogen responsible for gray (or grey) mold diseases. It has a broad host range, affecting many important agricultural crops. First described by Christiaan Hendrik Persoon in 1794 [1], it is considered a species within *Botrytis*, as it is the major pathogen within the genus [2]. *Botryotinia* spp. and *Botrytis* spp. previously caused taxonomy confusion because they induce similar disease symptoms as *Sclerotinia* spp. In 1949, Gregory confirmed that *Botryotinia fuckeliana* was the apothecial (sexual) stage of *B. cinerea*, establishing *Botryotinia* spp. and *Botrytis* spp. as anamorphs and teleomorphs of the same fungus [3].

*B. cinerea* belongs to the *Sclerotiniaceae* family of the class Leotiomycetes [4]. The *Botrytis* genus as to date comprises of 32 species [5]. Members of *Botrytis* genus are generally necrotrophic pathogens; they induce host cell death and lysis to access cellular nutrients. An exception to this is *B. deweyae*, which grows almost asymptomatically within *Hemerocallis* host species as an endophyte [6]. *Botrytis* species usually have a narrow host range, many are considered specialist pathogens that are restricted to infecting a single species or a small group of related species [7]. However, *B. cinerea* and *B. pseudocinerea* are generalists, with *B. cinerea* causing grey mold diseases in over 200 plants species [8]. Furthermore, *B. cinerea* is considered of higher agricultural and scientific importance due to its tendency of developing fungicide resistance [9].

*B. cinerea* is a highly successful pathogen due to its flexible infection modes, high reproductive output, wide host range and ability to survive for extended periods as conidia and/or small hardened mycelia masses called sclerotia [8]. It is primarily airborne as asexual conidia spores from mature conidiophores serve as major means of transmission (Figure 1). Mycelia from sclerotia and other infected tissues as well as seeds serve as alternative inoculum. In addition to its asexual cycle, sclerotia of *B. cinerea* can undergo a heterothallic sexual cycle to form apothecia, which release ascospores [8,9]. Production of ascospores from apothecia involves the fertilization of sclerotia with microconidia from an opposite mating type. However, the *B. cinerea* sexual cycle is rarely observed in nature [8].

Invasion of host plants by *B. cinerea* can start from damaged tissues or natural openings, allowing the fungus to establish infection [10]. Initial infected tissue often results in limited damage. However, massive conidia production enables fast subsequent spreading. For fruiting plants, the sites of infection are usually the floral organs and the fungus has diverse means of infecting different species. In grapes, conidia infect and spread in the receptacle area [11]. In raspberries and strawberries, the stigmatic fluid serves as adhesive and nutrient medium for the airborne conidia to germinate and develop hyphae [12]. Mycelia can then grow along pathway normally taken by pollen grains, thereby entering the carpel and invading the undeveloped ovaries.

After initial infection, *B. cinerea* enters a short phase where it exists as a biotroph within the plant [8]. Later on, it enters an aggressive necrotrophic phase, which is proposed to be triggered by biochemical changes such as an increase in volatile organic compounds, sugar and nitrogen contents in the ripening host tissues [13,14]. During this stage, the fungus secretes virulence factors such as oxalic acid (OA), cell wall degrading enzymes (CWDEs) and analogues of plant hormones to disrupt host metabolism, immune system and cellular structure [15,16]. Effects of virulence factors on hosts are characterized by fruit decay, resulting in softening of the flesh and a browning, leathery skin [17]. *B. cinerea* itself undergoes rapid growth of mycelia on plant surfaces and forms massive grey conidia masses [8]. At the same time, adjacent plant tissue may become infected, allowing for the spread of grey mold across the whole plant and to plants nearby. Upon killing the host tissue, *B. cinerea* can continue to grow saprophytically within the plant remains in the form of sclerotia and mycelia. The sclerotia of *B. cinerea* are highly resistant to environmental changes. They can remain viable in soil for up to 360 days, likely due to their melanized surface, β-glucan matrix and intracellular nutrient reserves such as protein, glycogen, polyphosphate and lipids [18,19].

Hosts of *B. cinerea* include over 500 species of mostly dicotyledons and some monocotyledons [8], many are of economic importance. Most severely affected are agricultural crops include vegetables (e.g., cucumber, tomato, zucchini) and fruit bearing plants (e.g., strawberry, grape, raspberry) [10,20]. It is estimated that *B. cinerea* causes a $10 to 100 billion of produce loss annually worldwide [21]. As an example, in Florida, BFR (botrytis fruit rot) contributed to a 36% decline in strawberry harvest from 2007 to 2016, which resulted in a net production value loss of $250 million annually [22]. Due to the highly destructive nature of *B. cinerea*, it was ranked second on a list of fungal pathogens of scientific and economic importance [23].

As with other fungal pathogens, the most common method of controlling *B. cinerea* spread is by chemical means; approximately 8% of the global fungicide market is used to control this pathogen [24]. However, fungicide usage is harmful to both the environment and human health [25]. Worse still, fungicide resistance in *B. cinerea* can develop quickly in the field. For example, resistance to dicarboximides arose after being used to prevent grey mold. Nowadays, the effectiveness of dicarboximides has severely decreased and it is only effective when used in combination with other fungicides [26]. As an alternative to chemical control, biological control via antagonistic yeasts has been used to inhibit the onset of grey mold post-harvesting, although the effectiveness of such method is often inconsistent. Cultural control is another viable method of minimizing fruit rot damage. Removal of excessive shoots and leaves reduces the formation and spread of sclerotia and/or conidia [27]. However, cultural control is often unrealistic for large scale commercial farming.

As with most major crop pathogens, the use of resistant cultivars is the most environmentally friendly and socially accepted control method. Some strong heritable host resistance sources against *B. cinerea* have been found, for example, in *Vitis* spp. (grapevine species). However, these resistant cultivars carry undesirable commercial traits such as thicker skins and increased epidermal waxes [28]. Similar to other necrotrophic pathogens, resistance against *B. cinerea* is mostly multigenic. In addition, fungal pathogen resistance genes appear to be negatively correlated with fungicide resistance genes, hindering efficient breeding [28].

In this review, we will focus on recent molecular findings from genes of *B. cinerea* that have been studied by mutant analysis. A brief summary on its genome will be discussed first. The *B. cinerea* genome has been sequenced recently and only been available for the last two decades [9]. It was updated in 2016 with improved coverage and annotation [29]. We hope this review aids in a comprehensive understanding of the molecular mechanisms of the biological processes behind this widespread airborne pathogen.

## 2. The features of *B. cinerea* Genome

### 2.1. Genomic Sequences

The *B. cinerea* strain B05.10 genome, originally isolated from grapes in California USA [30], was first sequenced in 2005 by Broad Institute and Syngenta Biotechnology using Sanger technology. Later Arachne assembled the genome in 2011 with 4.5x genomic coverage [9]. The most recent revision in 2016 produced a near-complete and gapless genome sequence using third-generation sequencing methods with data acquired by Illumina and PacBio, resulting in an increased 90x coverage [29]. The total *B. cinerea* genome was estimated to be 41.2 Mb in size (BioProject Accession number: PRJNA15632). It comprises of 18 chromosomes with an average 42.75% GC content [29]. After the removal of repetitive sequences, 10,701 protein-coding genes were predicted using Augustus and manual curation [29]. Another reference genome available is the *B. cinerea* strain T4 with a genome size of 41.6 Mb and 10,427 protein-coding genes [31]. However, we will be primarily focusing on the strain B05.10 genome analysis as it is the most frequently used strain in *B. cinerea* mutant analysis studies. Genes with strain-specific effects are specified.

Initial analysis of the *B. cinerea* strain B05.10 genome sequenced in 2005 unveiled numerous pathogenicity-related genes [9], including key genes involved in reactive oxygen species (ROS) generation and tolerance, peptidase secretion, OA biosynthesis and genes encoding CWDEs. Furthermore, significant number of genes involved in secondary metabolism pathways were identified. *B. cinerea* genome contains 43 essential secondary metabolites (SM) enzyme-encoding genes, 24 of which are *B. cinerea* specific. As a result, *B. cinerea* can potentially produce over 40 different SM, including sesquiterpene botrydial and polyketide botcinic acid phytotoxins [32]. Interestingly, an intein element encoding a homing endonuclease (HEG) was found within the *Prp8* gene in *B. cinerea* strain B05.10, which might have been acquired by horizontal gene transfer [33]. During meiosis with intein +/- heterozygotes, the HEG induces gene conversion between intein-carrying and empty *Prp8* isolates, whereby the HEG copies the intein into the empty allele regardless of independent assortment. Later annotation of the carbohydrate-active enzymes (CAZymes) revealed a wide diversity of 229 different CAZymes, likely responsible for the decomposition of pectin, hemicellulose and cellulose in plant cell walls [34]. The large variety of CAZymes implies their critical roles in defining the broad host range of *B. cinerea*.

In addition to whole genome sequencing *of B. cinerea* strain B05.10 to a gapless near-finish state in 2016, an optical map and a genetic linkage map were constructed to minimize assembly errors [29]. This gapless genome has become the standardized *B. cinerea* genome due to the vast improvements in accuracy and completion over other genome versions. It also features the first sequencing of mini-chromosomes 17 and 18, containing 18 and 14 genes respectively [29]. In addition, 40 regions on chromosomes with high reshuffling rates were identified, suggesting a high frequency of recombination in *B. cinerea* during sexual reproduction.

In 2015, Atwell et al. sequenced the genomes of 13 *B. cinerea* field isolates collected from Germany, United Kingdom, California and South Africa to determine genetic variation within the species [35]. Consensus sequences in genomes were aligned and alleles variants were annotated with pairwise single nucleotide polymorphisms (SNPs) analysis. Conceivably, the most common mutations found were insertions, deletions and point mutations; an average of 28 polymorphisms per kb were found within the *B. cinerea* population. Intriguingly, the genomic diversity within *B. cinerea* seems higher relative to that in other fungal species. Further analysis on genomic variation revealed a significant number of recombinant breakpoints, suggesting extensive whole genome recombination among *B. cinerea* strains during meiosis. Lab propagated *B. cinerea* isolates were also sequenced by Atwell et al. Contrary to the mutation accumulation hypothesis [36], SNPs differences across generations cannot be attributed to elevated mutation rates [35]. This suggests that the broad host range and rapid fungicide resistance development in *B. cinerea* are likely a result of the high genetic diversity and recombinant shuffling within the *B. cinerea* population rather than spontaneous mutations. The sexual cycle of the pathogen likely occurs more prevalently in nature than under laboratory conditions, contributing to its versatile adaptivity.

### 2.2. Transcriptomic and Secretomic Analysis

The availability of the whole genome sequences has allowed transcriptomic and secretomic analysis of *B. cinerea*, permitting future investigations into biological mechanisms. Changes in gene expression between stages of conidia development were identified using microarray focusing on the initial 15 h of infection; specifically stages of dormancy, pregermination, postgermination, appressoria development and early mycelia growth [37]. These data provided evidence that a large alteration in gene expression during conidial germination aids in host cell invasion by germ tube outgrowth and appressorium differentiation.

In recent years, RNA-Seq has become a popular and flexible method to identify genes potentially associated with certain biological processes. Many transcriptomic analyses were carried out to identify contributing factors of virulence in *B. cinerea.* For example, a recent co-transcriptomic study on plant-necrotroph interactions between the host *Arabidopsis thaliana* and pathogen *B. cinerea* [38] discovered 10 novel pathogen co-expression networks (GCNs) encoding potential virulence factors, including proteins involved in SM secretion, copper acquisition and detoxification. The co-transcriptome data also revealed that *B. cinerea* GCNs are negatively associated with plant immunity and positively associated with plant photosynthetic potential. These newly identified GCNs provide a future research avenue for reverse genetic analysis to determine the molecular mechanisms of plant-host interactions and pathogenesis in *B. cinerea*.

Secretomics is another useful method for identifying key genes by analyzing secreted proteins and their secretion pathways. One of the previous secretomic researches on *B. cinerea* predicted 34.38% of all gene products to be extracellular proteins secreted by the classic endoplasmic reticulum (ER)-Golgi secretion pathway [39]. Types of secreted proteins include CAZymes, proteases, proteins activating host plant hypersensitive response (HR) and other enzymes involved in oxidative burst, OA biosynthesis. Environmental conditions such as nutrients, ambient pH and metal ions influences protein secretion levels and composition [40]. Combining secretomics and transcriptomics datasets, these newly identified GCNs and secreted proteins provide a rich reservoir of candidates for reverse genetic analysis to determine the molecular mechanisms of plant-host interactions and pathogenesis in *B. cinerea*.

Mutant analysis is essential to establish a causal relationship between a gene and the affiliated biological process, beyond the association seen in omics analysis. Therefore, the rest of this review will provide an overview of all *B. cinerea* genes studied to date with mutant analysis methods (Table 1). A Venn diagram of the encoded proteins has been provided for readers’ overview (Figure 2). It is clear to see from the diagram that the majority of genes studied so far are either virulence-specific or associated with both virulence and development. A chromosomal map is constructed to detect possible clustering of virulence-associated genes (Figure 3). However, no significant gene clustering can be observed, likely because *B. cinerea* is still an understudied organism regardless of its economic importance. We also include a cell signaling diagram summarizing major signaling pathways discussed in the review (Figure 4).

## 3. Molecular Dissection of *B. cinerea* Biology

There are many gene nomenclature variations used by *Botrytis* researchers. To avoid confusion, we will be adopting the most commonly used gene/mutant/protein nomenclature. For example, ABC1 protein is encoded by italicized wild-type (WT) gene *ABC1*, while the mutant is designated by lowercase *abc1*. Exceptions will be indicated. It should be kept in mind of the different source strains of *B. cinerea* and plant hosts used in different studies (Table 1), which may contribute to some discrepancies in observed mutant phenotypes of the same genes.

### 3.1. Hyphal Growth and Virulence

Over the last two decades, accessibility to the *B. cinerea* whole genome sequence and advancements in mutant analysis techniques have greatly improved molecular studies of this fungus [29]. Most of the growth-related genes studied so far are highly conserved genes. Therefore, it is not surprising that these mutants often exhibit defects in sclerotial formation and/or conidiation (Table 1). They will be discussed in following sections. In this section, only genes specifically affecting hyphal growth and virulence but not differentiation, will be discussed. As mycelial growth precedes formation of structures such as appressoria, sclerotia and conidiophores, mutants with reduced growth likely would manifest diminished virulence as well. However, there are exceptions such as mutants *sho1* and *sln1*, which exhibit severely defective growth, sclerotial formation and conidiation but normal virulence (To be discussed later).

The single and double knockout mutants of highly conserved calcium channel proteins copper chaperone (CCH1) and mating pheromone-induced death (MID1) led to reduced vegetative growth under low environmental calcium conditions [41]. Other functions such as differentiation and pathogenicity were not affected. The aforementioned channel proteins facilitate intracellular calcium homeostasis through the Ca^2+^/Calcineurin (CN) signaling system. Although the similar phenotypes between *cch1* and *mid1* mutants suggested a complex formation by CCH1 and MID1, no experimental evidence of any interactions was found in *B. cinerea*. In mutants of other fungal species, growth under low calcium conditions is always impacted, along with virulence and development also being affected to varying degrees [42,43]. This suggests that *CCH1* and *MID1* play alternative roles in later stages of signaling pathways unlike in other fungi but they are universally essential to maintain calcium homeostasis for optimal growth.

*BCG1* and *BCG2* encode *B. cinerea* G alpha (Gα) subunits, which are components of heterotrimeric GTP-binding proteins [44]. They transduce environmental signals to activate many signaling cascades such as the well-known cyclic adenosine monophosphate (cAMP) pathway and Ca^2+^/CN signaling system. Previous studies indicated that Gα proteins are critical for conidiation and appressorium formation [45]. Deletion of *BCG1* in *B. cinerea* caused a decrease in growth rate and compacted colonies. Curiously, exogenous application of cAMP rescued the colony morphology, suggesting BCG1 is a direct or indirect activator of adenylyl cyclase and the cAMP-dependent pathway (discussed below) [44]. Although *bcg1* mutants were able to initially colonize and penetrate host tissues, further host invasion and spread did not occur. The *bcg1* mutants also failed to secrete proteases, suggesting BCG1 contributes to secretion of hydrolytic enzymes. BCG2 belongs to subfamily II of fungal Gα proteins, whose members are not well studied [44]. Unlike *bcg1*, *bcg2* mutants undergo normal growth and infection but with slower lesion expansion. Though further research is still needed to fully understand the signaling pathway and effects of BCG2, these data suggest that BCG1 and BCG2 may control different signal transduction pathways.

**Table 1 pathogens-09-00923-t001:** Summary of genes from *B. cinerea* that have been studied using mutant analysis.

Gene Code (New)	Fungal Strains	Mutant Name	Gene Full Name	Mutant Type	Mutant Phenotypes								Host Species	Other Functions of Encoded Protein	Reference
					Hyphal Growth	Sclerotial Formation	Oxalate Production	Virulence	Compound Appressoria Formation (Penetration)	Conidiation/Sporulation	Induce Host HR/Resistance	Secretion Signal			
Bcin01g00550	B05.10 strain	*sas1*	secretion-related Rab/GTPase gene	deletion	+	+	NA	+	NA	+	NA	No		Protein secretion, sporulation	[46]
Bcin01g02000	B05.10 strain	*rac*	Small GTPases	deletion	+	+	NA	+	NA	+	NA	No		Polar growth, reproduction	[47]
Bcin01g02730	B05.10 strain	*vel2*	velvet-like gene	deletion	+	+	+	+	NA	+	NA	No		Light response, acidification	[48]
Bcin01g02730	38B1 strain	*velB*	velvet-like gene	deletion	+	+	NA	+	-	+	NA	No		Negative role in asexual development and melanin biosynthesis	[49]
Bcin01g02790	Chickpea isolate from fields of Govind Ballabh Pant University	*dgat2*	diacylglycerol O-acyl transferase 2	T-DNA, deletion	-	+	+	+	NA	+	NA	No		Penetration and consequently virulence	[50]
Bcin01g03790	Bd90 strain	*chs4*	chitin synthases	deletion	-	-	NA	-	NA	-	NA	No			[51]
Bcin01g06080	38B1 strain	*pro40*	scaffold protein	deletion	NA	NA	+	NA	NA	NA	NA	No			[52]
Bcin01g07770	B05.10 strain	*trx2*	thioredoxin	deletion	-	NA	NA	-	-	-	NA	No		Resist to oxidative stress; strx1/trx2 double mutant has retarded growth as trr1	[53]
Bcin01g08050	B05.10 strain	*ass1*	argininosuccinate synthase	deletion	+	NA	NA	+	NA	NA	NA	No		Production of L-arginine	[54]
Bcin01g08230	B05.10 strain	*crz1*	calcineurin-Responsive Zinc Finger Transcription Factor	deletion	+	+	NA	+	NA	+	NA	No		Acts downstream of calcineurin but not the only target of calcineurin	[55]
Bcin01g08690	B05.10 strain	*noxD*	component of the NADPH oxidase complex	deletion	NA	+	NA	+	-	+	NA	No		Interact with NOXA	[56]
Bcin01g09450	B05.10 strain	*lgd1*	galactonate dehydratase gene	deletion	-	NA	NA	+	NA	NA	defence-related genes were not induced	No	*Arabidopsis thaliana* and *Nicotiana benthamiana*, not *Solanum lycopersicum*	D-galacturonic acid catabolism	[57]
Bcin02g00190	B05.10 strain	*jar1*	Histone 3 Lysine 4 (H3K4) demethylation	deletion	-	+	NA	+	+	+	NA	No		Oxidative and low-oxygen stress adaptation	[58]
Bcin02g01540	B05.10 strain	*cnA*	catalytic subunit of calcineurin	deletion	+	+	NA	+	NA	+	NA	No			[59]
Bcin02g02570	B05.10 strain	*atg8*	autophagy-related gene	deletion	+	+	NA	+	NA	+	NA	No		Interact with ATG4, lipid droplet metabolism	[60]
Bcin02g04360	B05.10 strain	*ygh1*	alpha/beta hydrolases	deletion (heterokaryotic)	+	+	NA	-	NA	+	NA	No		Required for the formation of the key intermediate T4HN	[61]
Bcin02g04930	B05.10 strain	*noxB*	NADPH oxidases	deletion	-	+	-	+	+	-	NA	No		Penetration	[62]
Bcin02g06470	B05.10 strain	*str2*	cystathionine γ-synthase	deletion	+	+	NA	+	NA	+	NA	No		Response to various stresses	[63]
Bcin02g06590	38B1 strain	*bck1*	MAPK cascade	deletion	+	NA	-	+	+	+	NA	No		Melanin biosynthesis	[52]
Bcin02g06770	B05.10 strain	*atg4*	cysteine protease	deletion	+	+	NA	+	+	+	NA	No		Autophagy	[64]
Bcin02g07700	IK2018/B05.10	*ara1*	α-1,5-L-endo-arabinanase	deletion	-	NA	NA	+	NA	NA	NA	Yes	*Arabidopsis thaliana*	Host dependent, secondary lesion formation during infection	[65]
Bcin02g07970	B05.10 strain	*bhp1*	hydrophobin encoding gene	deletion	-	-	NA	-	-	-	NA	Yes		Development of apothecia	[66,67]
Bcin02g08570	B05.10 strain	*sec14*	protein secretion related gene	deletion	-	NA	NA	+	NA	-	NA	No		Protein secretion	[68]
Bcin02g08650	38B1 strain	*skn7*	response regulator in the high-osmolarity glycerol pathway	deletion	-	+	NA	-	NA	+	NA	No		Regulation of vegetative differentiation and in the response to various stresses	[69]
Bcin02g08760	B05.10 strain	*smr1*	sclerotial melanogenesis-regulatory gene	deletion	NA	+	NA	NA	NA	NA	NA	No		Sclerotial melanogenesis	[70]
Bcin02g08770	B05.10 strain	*pks12*	polyketide synthase	deletion	-	+	NA	-	NA	-	NA	No			[61]
Bcin03g00500	B05.10 strain	*spl1*	cerato-platanin family protein	deletion	-	NA	NA	+	NA	NA	+	Yes	a variety of hosts	HR and PR gene induction, BAK1 required	[71]
Bcin03g01490	B05.10 strain	*lga1*	keto-3-deoxy-L-galactonate aldolase gene	deletion	-	NA	NA	+	NA	NA	defence-related genes were not induced	No	*Arabidopsis thaliana* and *Nicotiana benthamiana*, not *Solanum lycopersicum*	D-galacturonic acid catabolism	[57]
Bcin03g01500	B05.10 strain	*gar2*	galacturonate reductase genes	deletion	-	NA	NA	+	NA	NA	defence-related genes were not induced	No	*Arabidopsis thaliana* and *Nicotiana benthamiana*, not *Solanum lycopersicum*	D-galacturonic acid catabolism	[57]
Bcin03g01720	38B1 strain	*ptc1*	Type 2C Ser/Thr phosphatases	deletion	+	+	NA	+	NA	+	NA	No		Melanin biosynthesis, regulation of multiple stress tolerance and virulence	[72]
Bcin03g02380	B05.10 strain	*mid1*	calcium channel protein	deletion	+	-	NA	-	-	-	NA	No		Vegetative growth under conditions of low extracellular calcium	[41]
Bcin03g02930	B05.10 strain	*cla4*	Rac effectors	deletion	+	NA	NA	+	NA	+	NA	No		Cell cycle regulating processes downstream of RAC	[73]
Bcin03g03060	B05.10 strain	*bcdh*	UDP-glucose-4,6-dehydratase	deletion	-	-	NA	-	NA	-	NA	No		Production of rhamnose-containing glycan	[74]
Bcin03g03390	B05.10 strain	*sod1*	Cu-Zn-superoxide dismutase	deletion	NA	NA	NA	+	NA	NA	NA	No	*Phaseolus vulgaris*		[75]
Bcin03g05410	collected from chickpea feld of Govind Ballabh Pant University	*klp7*	kinesin	T-DNA	+	NA	+	+	+	+	NA	No		Reduced activities of polygalacturonase (PG) and pectin methyl esterases (PME)	[76]
Bcin03g06840	B05.10 strain	*noxR*	regulatory subunit of the Nox complex	deletion	-	+	-	+	+	-	NA	No		Activation of both NOX enzymes	[62]
Bcin03g06910	B05.10 strain	*yak1*	dual-specificity tyrosine phosphorylation-regulated protein kinase	deletion	-	+	NA	+	+	+	NA	No		Adaptation to oxidative stress and triadimefon	[77]
Bcin03g07190	38B1 strain	*mkk1*	MAPK kinase	deletion	+	NA	+	+	+	+	NA	No		Melanin biosynthesis, negatively regulates oxalic acid biosynthesis	[52]
Bcin03g07420	B05.10 strain	*reg1*	ortholog of the F. oxysporum transcriptional regulator FoSge1	deletion	-	-	NA	+	-	+	NA	No	*Phaseolus vulgaris*	Toxin production	[78]
Bcin03g07900	B05.10 strain	*exo70*	exocyst subunit gene	deletion	+	+	NA	+	NA	+	NA	No			[79]
Bcin03g08050	B05.10 strain	*pks13*	polyketide synthase	deletion	+	+	+	+	NA	+	NA	No		Melanin synthesis, mutant shows enhanced growth rate and virulence, white sclerotia	[80]
Bcin03g08100	B05.10 strain	*brn2*	tetrahydroxynaphthalene (THN) reductases	deletion	-	+	NA	-	NA	+	NA	No			[61]
Bcin03g08110	B05.10 strain	*scd1*	scytalone dehydratases	deletion	-	+	NA	-	NA	+	NA	No			[61]
Bcin04g00340	38B1 strain	*ptc3*	Type 2C Ser/Thr phosphatases	deletion	+	+	NA	+	NA	+	NA	No		Melanin biosynthesis, regulation of multiple stress tolerance and virulence	[72]
Bcin04g01630	B05.10 strain	*pkaR*	regulatory regulatory subunit of cAMP-dependent protein kinase	deletion	+	-	-	+	NA	-	NA	No			[81]
Bcin04g03140	B05.10 strain	*ras2*	fungal-specific Ras GTPase	deletion	+	-	-	+	NA	-	NA	No		Conidial germination	[81]
Bcin04g04800	B05.10 strain	*brn1*	tetrahydroxynaphthalene (THN) reductases	deletion	+	+	+	+	NA	+	NA	No		Melanin synthesis, mutant shows enhanced growth rate and virulence, orange sclerotia	[80]
Bcin04g05300	B05.10 strain	*glr1*	GSH reductase	deletion	-	NA	NA	+	+	NA	NA	No		Conidia germination	[53]
Bcin04g05920	B05.10 strain	*sep4*	septin gene	deletion	-	+	NA	+	+	+	NA	No		Melanin and chitin accumulation in hyphal tips	[82]
Bcin05g00240	HYOGO11	*ccc2*	copper-transporting ATPase	deletion	-	+	NA	+	+	+	NA	No		Melanization	[83]
Bcin05g00350	B05.10 strain	*noxA*	NADPH oxidases	deletion	-	+	-	+	-	-	NA	No		Colonize the host tissue	[62]
Bcin05g00760	B05.10 strain	*cdc24*	GEF (guanine nucleotide exchange factor)	deletion (heterokaryotic)	+	NA	NA	+	+	+	NA	No			[84]
Bcin05g01210	B05.10 strain	*lae1*	putative interaction partner of BcVEL1	deletion	-	+	+	+	NA	+	NA	No		Putative interaction partner of VEL1	[53]
Bcin05g01430	B05.10 strain	*glr2*	GSH reductase	deletion	-	NA	NA	-	-	NA	NA	No			[53]
Bcin05g02680	B05.10 strain	*trr1*	thioredoxin reductase	deletion	+	NA	NA	+	-	-	NA	No		Resist to oxidative stress	[53]
Bcin05g04030	B05.10 strain	*mads1*	MADS-box transcription factor	deletion	+	+	NA	+	NA	+	NA	No		Regulates the expression of light-responsive genes	[68]
Bcin05g06320	T4 strain	*bcp1*	Cyclophilin A	deletion	-	NA	NA	+	NA	-	NA	No			[85]
Bcin05g06770	B05.10 strain	*bcg1*	Gαi subunits (I)	deletion	+	NA	NA	+	-	NA	NA	No		Protease secretion	[44]
Bcin05g08290	B05.10 strain	*iqg1*	fungal homolog of the RasGAP scaffold protein IQGAP	deletion	_	+	NA	+	+	+	NA	No		Resistance against oxidative and membrane stress	[86]
Bcin06g00026	B05.10 strain	*mfsG*	Major Facilitator Superfamily transporter	deletion	NA	NA	NA	+	NA	NA	NA	No		Increases tolerance to glucosinolates	[87]
Bcin06g00240	B05.10 strain	*hbf1*	hyphal branching-related factor 1	T-DNA, deletion	-	+	NA	+	+	-	NA	No		Hyphal branching	[88]
Bcin06g00450	B05.10 strain	*bhl1*	Botrytis hydrophobin-like gene	deletion	-	-	NA	-	-	-	NA	Yes			[67]
Bcin06g00510	B05.10 strain	*bhp3*	hydrophobin encoding gene	deletion	-	-	NA	-	-	-	NA	Yes		Development of apothecia	[66,67]
Bcin06g02380	B05.10 strain	*bcer*	UDP-4-keto-6-deoxyglucose-3,5-epimerase/-4-reductas	deletion	+	+	NA	+	NA	+	NA	No		Production of rhamnose-containing glycan	[74]
Bcin06g03440	B05.10 strain	*aox*	alternative oxidase	deletion	-	+	NA	+	NA	+	NA	No		Adaptation to environmental stress	[89]
Bcin06g03990	B05.10 strain	*ku70*	inhibitor of NHEJ	deletion	-	NA	NA	-	NA	-	NA	No		Ku deficiencies improved HR efficiency	[90]
Bcin06g04390	B05.10 strain	*rho3*	small GTPases of the Rho family	deletion	+	+	NA	+	+	+	NA	No			[91]
Bcin06g04660	B05.10 strain	*gar1*	galacturonate reductase genes	deletion	-	NA	NA	+	NA	NA	defence-related genes were not induced	No	*Arabidopsis thaliana* and *Nicotiana benthamiana*, not *Solanum lycopersicum*	D-galacturonic acid catabolism	[57]
Bcin06g06040	B05.10 strain	*sun1*	Group-I SUN family of proteins	deletion	-	+	NA	+	NA	+	NA	Yes		Production of reproductive structures and adhesion to plant surface	[92]
Bcin06g07300	B05.10 strain	*mtg2*	Obg protein	deletion	+	+	NA	+	NA	+	NA	No		Asexual development, environmental stress response	[30]
Bcin07g00720	B05.10 strain	*atg1*	autophagy-related gene	deletion	+	+	NA	+	+	+	NA	No		Lipid metabolism	[93]
Bcin07g01300	Bd90 strain	*chs7*	chitin synthases	deletion	-	-	NA	+	NA	-	NA	No	*Phaseolus vulgaris*, ecotype Col-0 of *Arabidopsis thaliana*	Virulence depends on host plants	[51]
Bcin07g02480	B05.10 strain	*pmr1*	P-type Ca2+/Mn2+-ATPase	deletion	+	+	NA	+	NA	+	NA	No	*Solanum lycopersicum* leaves and fruit and *Malus domestica* fruit	Biofilm formation	[94]
Bcin07g02610	B05.10 strain	*pgd*	6-phosphogluconate dehydrogenase	deletion	+	NA	NA	+	NA	+	NA	No		Influenced by NOX	[95]
Bcin07g03050	B05.10 strain	*kdm1*	histone 3 lysine 36 (H3K36)-specific demethylas	T-DNA, deletion	+	+	NA	+	NA	+	NA	No		Stress responses and photomorphogenesis	[96]
Bcin07g03340	B05.10 strain	*nma*	high-temperature requirement (HtrA) family of serine proteases	deletion	-	NA	NA	-	NA	-	NA	No		Pro-apoptotic activity	[97]
Bcin07g05880	B05.10 strain	*vel3*	velvet-like gene	deletion	-	-	-	-	NA	+	NA	No		Light response, acidification	[48]
Bcin08g00120	B05.10 strain	*aqp8*	aquaporin 8	deletion	+	+	NA	+	+	+	NA	No		Pigment metabolism	[98]
Bcin08g00550	B05.10 strain	*pde2*	phosphodiesterase	deletion	+	+	NA	+	NA	+	NA	No		Camp signaling pathway	[99]
Bcin08g00850	38B1 strain	*ptpB*	putative protein tyrosine phosphatase (PTP) gene	deletion	+	+	NA	+	NA	+	NA	No		Negative role in melanin biosynthesis; PTPA and PTPB have opposite functions in conidiation	[100]
Bcin08g01740	38B1 strain	*brrg1*	putative response regulator protein	deletion	-	NA	NA	-	NA	+	NA	No		Sensitivity to fungicides and osmotic stress	[101]
Bcin08g02970	Bd90 strain	*pme1*	pectin methylesterase	deletion	-	-	NA	+	NA	-	NA	Yes			[102]
Bcin08g02990	B05.10 strain	*ser2*	subtilisin-like protease 2	deletion	+	+	NA	+	+	+	NA	Yes			[103]
Bcin08g03910	B05.10 strain	*pka2*	catalytic subunit of cAMP-dependent protein kinase	deletion	-	-	-	-	NA	-	NA	No			[81]
Bcin08g04530	B05.10 strain	*atg3*	ubiquitin-like (UBL) protein-activating enzymes	deletion	+	+	NA	+	NA	+	NA	No		Autophagy	[104]
Bcin08g05150	B05.10 strain	*sho1*	biosensors of HOG pathway	deletion	+	+	NA	-	NA	+	NA	No		Redundant for sln1 mutant	[105]
Bcin08g06620	B05.10/T4strain	*hox8*	homeobox transcription factor encoding gene	deletion	+	-	NA	+	+	+	NA	No			[106]
Bcin09g01800	B05.10 strain	*xyl1*	Xylanase	deletion	-	NA	NA	+	NA	NA	+	Yes		Trigger PTI	[107]
Bcin09g02390	B05.10 strain	*bmp3*	cell wall integrity MAPK	deletion	+	+	NA	+	+	+	NA	No		Melanin biosynthesis	[108]
Bcin09g02820	B05.10 strain	*rcn1*	calcipressin	deletion	+	-	NA	+	NA	-	NA	No		Positive modulator of CNA	[59]
Bcin09g03710	B05.10 strain	*frp1*	FRP1 F-box gene	deletion	-	+	NA	-	NA	-	NA	No		Sexual reproduction	[109]
Bcin09g04170	B05.10 strain	*glk1*	glucokinase	deletion	-	NA	NA	-	NA	-	NA	No			[110]
Bcin09g04730	B05.10 strain	*atg7*	ubiquitin-like (UBL) protein-activating enzymes	deletion	+	+	NA	+	NA	+	NA	No		Autophagy	[104]
Bcin09g06130	B05.10 strain	*pls1*	tetraspanin	deletion	-	-	-	+	+	-	NA	No		Penetration; sexual development	[111]
Bcin09g06880	B05.10 strain	*lip1*	lipase gene	deletion	NA	NA	NA	-	NA	NA	NA	Yes		Catabolite repression	[112]
Bcin09g06900	38B1 strain	*bos5*	mitogen-activated protein kinase kinase	deletion	+	NA	NA	+	-	+	NA	No		Adaptation to iprodione and ionic stress	[113]
Bcin10g00450	B05.10 strain	*pde1*	phosphodiesterase	deletion	-	-	NA	-	NA	-	NA	No		Camp signaling pathway, enhance PDE2 function	[99]
Bcin10g01250	B05.10 strain	*bag1*	Bcl-2 associated athanogene	deletion	+	+	NA	+	+	+	NA	No		Hyphal melanization, response to multiple abiotic stresses and UPR pathway	[114]
Bcin10g02180	B05.10 strain	*cfem1*	CFEM protein with putative GPIanchored site	deletion	-	-	NA	+	NA	+	NA	Yes		Stress tolerance	[115]
Bcin10g02530	B05.10 strain	*ser1*	subtilisin-like protease 1	deletion	-	-	NA	-	-	-	NA	Yes			[103]
Bcin10g05490	B05.10 strain	*ku80*	inhibitor of NHEJ	deletion	-	NA	NA	-	NA	-	NA	No		Ku deficiencies improved HR efficiency	[90]
Bcin10g05950	B05.10 strain	*pacC*	PacC transcription factor	deletion	+	+	+	+	NA	+	+	No		Production of reactive oxygen species; enzyme secretion	[116]
Bcin11g01450	B05.10 strain	*bhp2*	hydrophobin encoding gene	deletion	-	-	NA	-	-	-	NA	Yes		Development of apothecia	[66,67]
Bcin11g01720	B05.10 strain	*ltf3*	putative C2H2 transcription factor	deletion	-	NA	NA	-	NA	+	NA	No			[78]
Bcin11g02360	B05.10 strain	*dim5*	Histone H3 Lysine 9 Methyltransferase	deletion	+	+	NA	+	NA	+	NA	No			[117]
Bcin11g03560	38B1 strain	*os4*	mitogen-activated protein kinase kinase kinase gene	deletion	+	NA	NA	+	-	+	NA	No		Adaption to hyperosmotic and oxidative stresses	[118]
Bcin11g05350	B05.10 strain	*mctA*	putative monocarboxylate transporter	deletion	-	+	NA	+	NA	+	NA	No		Pyruvate uptake	[119]
Bcin11g05700	B05.10 strain	*hxk1*	hexokinase	deletion	+	NA	NA	+	NA	+	NA	No	fruit	Sugar metabolism	[110]
Bcin11g05810	B05.10 strain	*cch1*	calcium channel protein	deletion	+	-	NA	-	-	-	NA	No		Vegetative growth under conditions of low extracellular calcium	[41]
Bcin12g01360	B05.10 strain	*fkbp12*	FK506-binding protein	deletion	-	NA	NA	-	NA	-	NA	No		Sulfur repression of the synthesis of a secreted serine protease	[120]
Bcin12g02530	B05.10 strain	*bem1*	scaffold protein	deletion	-	NA	NA	+	+	+	NA	No		Part of a polarity complex involving the GEF CDC24	[84]
Bcin12g02750	B05.10 strain	*elp4*	elongator complex protein	deletion	+	+	NA	+	NA	NA	NA	No		Mycelia differentiation, melanization, various environmental stress response	[121]
Bcin12g03770	B05.10 strain	*nop53*	pre-rRNA processing factor	deletion	+	+	NA	+	+	+	NA	No		Oxidative and osmotic stress adaptation	[122]
Bcin12g03880	B05.10 strain	*pp2ac*	a catalytic subunit of a PP2A serine/threonine protein phosphatase	T-DNA, RNAi	+	+	NA	+	NA	-	NA	No		Resistance to H_2_O_2_	[123]
Bcin12g04280	B05.10 strain	*trx1*	thioredoxin	deletion	-	NA	NA	+	-	-	NA	No		Resist to oxdative stress; bstrx1bctrx2 double mutant has retarded growth as bctrr1	[53]
Bcin12g04900	BC22 strain	*kmo*	kynurenine 3-monooxygenase (KMO)	T-DNA	+	+	+	+	NA	+	NA	No		Cell wall degrading enzymes activity	[124]
Bcin12g05360	Bd90 strain	*chs6*	chitin synthases	deletion (heterokaryotic )	+	-	NA	+	NA	+	NA	No		Sexual cycle	[51]
Bcin12g05760	B05.10 strain	*ras1*	Small GTPases	deletion	+	+	NA	+	NA	+	NA	No		Polar growth, reproduction	[47]
Bcin12g06380	T4 strain	*bot1*	P450 monooxygenase	deletion	NA	NA	NA	+	NA	NA	NA	No		Strain-specific virulence factor	[125]
Bcin13g00090	B05.10 strain	*cdc42*	small GTPase	deletion	+	+	NA	+	+	+	NA	No		Conidial germination and nuclear distribution	[126]
Bcin13g05340	B05.10 strain	*hp1*	heterochromatin protein 1	deletion	-	-	NA	-	NA	-	NA	No			[117]
Bcin13g05610	B05.10 strain	*cpa1*	capping protein (CP) subunit	deletion	+	+	NA	+	+	+	NA	No		Conidial germination; interact with CPB1	[127]
Bcin14g00610	B05.10 strain	*pg2*	endopolygalacturonase enzyme	deletion	NA	NA	NA	+	NA	NA	NA	Yes	*Solanum lycopersicum* and *Vicia faba*		[128]
Bcin14g01730	B05.10 strain	*bcg2*	group II of Gα subunits	deletion	-	NA	NA	+	-	NA	NA	No			[44]
Bcin14g01870	B05.10 strain	*sln1*	biosensors of high-osmolarity glycerol(HOG) pathway	deletion	+	+	NA	-	NA	+	NA	No		Redundant for shn1 mutant	[105]
Bcin14g03930	B05.10 strain	*ltf1*	light-responsive transcription factor 1	deletion	+	NA	NA	+	+	+	NA	No		ROS homoeostasis, light-dependent differentiation	[129]
Bcin14g04650	B05.10 strain	*sec31*	protein secretion related gene	deletion	-	NA	NA	+	NA	-	NA	No		Protein secretion	[68]
Bcin14g05500	B05.10 strain	*god1*	putative glucose oxidase gene	deletion	NA	NA	NA	-	NA	NA	NA	Yes	*Phaseolus vulgaris*		[75]
Bcin15g00280	38B1 strain	*rim15*	Per-Arnt-Sim (PAS) kinase	deletion	+	NA	+	-	NA	NA	NA	No			[52]
Bcin15g00450	B05.10 strain	*dim2*	DNA methyltransferase	deletion	-	-	NA	-	NA	-	NA	No			[117]
Bcin15g01330	38B1 strain	*ptpA*	putative protein tyrosine phosphatase (PTP) gene	deletion	+	+	NA	+	NA	+	NA	No		Negative role in melanin biosynthesis; bcptpa and bcptpb have opposite functions in conidiation	[100]
Bcin15g02590	B05.10 strain	*bac*	adenylate cyclase	deletion	+	NA	NA	+	NA	+	NA	No			[130]
Bcin15g03390	38B1 strain	*veA*	velvet-like gene	deletion	-	+	NA	+	-	+	NA	No		Negative role in asexual development and melanin biosynthesis	[49]
Bcin15g03390	B05.10 strain	*vel1*	VELVET Gene	deletion	NA	+	+	+	NA	+	NA	No			[131]
Bcin15g03580	B05.10 strain	*sak1*	Hog-type stress-activated MAPK	deletion	NA	+	NA	+	NA	+	NA	No		Early stages of infection; regulation of secondary metabolism	[132]
Bcin15g04040	B05.10 strain	*spt3*	SPT3 subunit of a Spt-Ada-Gcn5-acetyltransferase	T-DNA, deletion	+	+	NA	+	NA	+	NA	No		Resistance to H_2_O_2_	[123]
Bcin15g04140	B05.10 strain	*bir1*	baculovirus IAP (inhibitor of apoptosis protein) repeat	partial knockout	+	NA	NA	+	NA	+	NA	No		Anti-apoptotic activity	[97]
Bcin16g00630	B05.10 strain	*pck1*	phosphoenolpyruvatecarboxykinase gene	T-DNA, deletion	-	+	NA	+	+	+	NA	No			[133]
Bcin16g01130	B05.10 strain	*pka1*	catalytic subunit of cAMP-dependent protein kinase	deletion	+	-	-	+	NA	-	NA	No			[81]
Bcin16g01780	B05.10 strain	*far1*	scaffold protein	deletion	-	NA	NA	-	-	-	NA	No		No obvious phenotypes	[84]
Bcin16g01820	B05.10 strain	*cgf1*	conidial germination-associated factor 1	deletion	-	-	NA	+	+	+	NA	No		ROS production, osmotic and oxidative stress adaptation	[134]
Bcin16g02020	B05.10 strain	*actA*	actin protein	deletion	+	NA	NA	+	NA	+	NA	No		Hyphae structure	[135]
Bcin16g04910	B05.10 strain	*sep1*	formin	T-DNA	+	NA	NA	+	+	+	NA	No		Septum formation and polarized growth	[84]
Bcin03g03480	B05.10 strain	*xyn10A*	xylanases of family GH10	RNAi	-	-	NA	+	NA	-	NA	Yes			[136]
Bcin05g06020		*xyn10B*	xylanases of family GH10									Yes			
Bcin03g00480		*xyn11A*	xylanases of family GH11									Yes			
Bcin15g01600		*xyn11B*	xylanases of family GH11									Yes			
Bcin12g00090		*xyn11C*	xylanases of family GH11									Yes			

The gene code is according to ASM185786v1 published in 2017. ‘+’ in the table represents that the phenotype of mutant is altered as compared to the corresponding WT strain, while ‘-’ indicates the mutant phenotype is unchanged. NA means not assessed. The secretion signals were found using ‘SignalP-5.0′. Unless specified, all deletions were generated with homologous recombination. The names of host species used for pathogenicity test of mutants are: *Arabidopsis thaliana* (Arabidopsis), *Nicotiana benthamiana* (tobacco), *Solanum lycopersicum* (tomato), *Phaseolus vulgaris* (French bean), *Malus domestica* (Apple), *Vicia faba* (Broad bean). The ecotype of *A. thaliana* mentioned in the table is Columbia-0.

The cAMP-dependent pathway requires protein kinase A (PKA) in order to phosphorylate downstream proteins under high cAMP concentration. PKA1 and PKA2 are catalytic subunits of PKA in *B. cinerea* [81]. Curiously, *pka1* mutants displayed retarded growth and virulence, while *pka2* mutants had no obvious phenotype change. Further studies on PKA activity indicated that PKA1 is the predominant catalytic subunit, as only *pka1* mutants have a detectable PKA activity decrease. PKAR is the regulatory subunit of PKA. Conceivably, deletion of *PKAR* displayed similar phenotypes and PKA activity as *pka1* mutants [81], suggesting that PKAR is essential for normal PKA1 function. However, the exact molecular mechanisms of PKAR and PKA1 interactions are not well established. Other studies in *Neurospora crassa* and *Cryphonectria parasitica* have suggested that PKA protein subunits regulate partner subunits transcription and degradation [137,138]. In *B. cinerea*, PKAR may likewise stabilizes PKA1 and is essential for maintaining normal PKA1 cellular concentration.

Deletion of argininosuccinate synthase gene *ASS1* resulted in a decrease in growth rate under insufficient extracellular L-arginine concentration [54]. A general decrease in virulence was also observed in most host tissues but infection still occurred in tissues containing high amount of L-arginine (e.g., grapes). However, lesion expansion was slowed down over time before halting as free L-arginine was consumed, suggesting that vegetative growth and initiation of pathogenic phase in *B. cinerea* are dependent on free arginine concentration.

### 3.2. Sclerotia Development

Sclerotia naturally form within dying host tissue. During early spring when warm and humid conditions occur, sclerotia can rapidly initiate conidiophores development and production [8]. Both mycelia and conidia can serve as inoculum to initiate new infection. When fertilized with microconidia during sexual cycle, apothecia can emerge from sclerotia and upon maturation, release ascospores (Figure 1). The sclerotia are typically pigmented black from 1,8-dihydroxynaphthalene (DNH) melanin, which aids in sclerotial survival in unfavorable environments [8]. Here we first discuss genes affecting only sclerotial structure and formation. The associated mutants developed defective sclerotia and often exhibit greatly reduced sclerotial survivability. Melanization associated genes, on the other hand, will be discussed later.

Botrytis-hydrophobin BHP1, BHP2 and BHP3 are small, nonpolar proteins responsible for coating fungal surfaces with hydrophobic layers. They serve many different functions in other fungi [66]. In *B. cinerea*, single mutants of *BHP1, BHP2* or *BHP3* did not have any aberrant phenotypes compared to WT. Double mutant *bhp1 bhp3* and triple mutant had an easily wettable sclerotia and suffered from compromised structural rigidity in humid conditions, suggesting BHP1 and BHP3 perform redundant functions in sclerotia development. Apothecia produced by fertilized sclerotia of double *bhp1 bhp2* and triple mutant resulted in swelling of apothecium structure and collapse upon outgrowth, indicating BHP1 and BHP2 play overlapping roles for such developmental process. When a hydrophobin-like protein BHL1 was also knocked out, however, no phenotypic changes from WT were observed [67]. It remains unknown what the role of BHL1 is and whether it shares redundancy with other BHPs in *B. cinerea*.

FRP1 (F-box protein required for pathogenicity) is a protein that is part of an SKP1, Cullin1 and F-box (SCF) complex E3 ligase, predicted to be involved in the ubiquitination of its substrate protein [139]. Its close homologs in *Fusarium oxysporum* are required for virulence and non-sugar carbon metabolism [140]. *FRP1* deletion in *B. cinerea* resulted in enhanced growth on simple sugars and inability to produce apothecia [109], indicating a function of FRP1 in suppressing growth and it is essential for apothecium development in *B. cinerea*. Sclerotium formation was delayed and the sclerotia formed were smaller, heavily pigmented and textured roughly. The *frp1* mutant phenotype variations across fungal species suggests that FRP1 may target different proteins in varied species, which will be interesting to examine in the future.

The following discussion will focus on genes affecting virulence, sclerotium formation and conidiation. In fungi, NADPH oxidases (NOX) homologs are involved in differentiation processes and ROS generation [62]. Single and double mutants of *noxA* and *noxB* led to failed formation of sclerotia and a reduction in conidiation and virulence [56,62]. Double *noxA noxB* mutant exhibited greater phenotypic changes than the sum of individual *noxA* and *noxB* changes, suggesting that NOXA and NOXB have overlapping biological functions. Virulence effects of *NOXA* are on lesion spread while *NOXB* affects host tissue penetration more. Interestingly, no change in ROS production and secretion was observed in either single or double mutants, indicating that NADPH oxidases in *B. cinerea* do not contribute significantly to the oxidative burst mechanism and are mainly involved in differentiation processes. The mutant *noxR*, with a defective regulatory NOX subunit, displayed similar phenotypes as double mutant *noxA noxB* [62,95], suggesting that NOXR is required for normal NOXA and NOXB function. In addition, a recent study revealed a new NADPH oxidases subunit, NOXD, whose mutant exhibited identical phenotype as *noxA* [56]. Localization and immunoprecipitation experiments suggested that NOXA requires interaction with NOXD for normal function and NOXD may be an activator or regulator of NOXA. Possible functions of NOXD include stabilizing NOXA or assisting in binding regulatory subunits such as NOXR.

IQ motif-containing GTPase-activating protein IQG1 serves as a linkage protein for multiple intracellular components in *B. cinerea*, including mitogen-activated protein kinase (MAPK), Ca^2+^/CN and the NOX complex. IQG1 homologs in mammalian cells are involved in a wide range of central signaling pathways. *iqg1* mutant is avirulent and produces significantly less conidia but more sclerotia [86]. The mutant phenotype is similar to those of *noxA* and *noxD* mutants, suggesting that IQG1 is required for normal NOXA and NOXD function. *IQG1* deletion hampers MAPK and Ca^2+^/CN pathway activation, suggesting IQG1 is a key component in linking multiple signaling systems in *B. cinerea*. IQG1 may act as a bridge resolving the lack of evidence for direct interactions between NOXA and NOXR. However, no supporting data have been available so far.

Protein kinases are critical to intracellular functions because of their ability to phosphorylate components of signal transduction systems to transduce signals [141]. Not surprisingly, protein kinases were found to be involved in pathogenicity and developmental processes. Mutation of the dual-specificity tyrosine kinase Yet Another Kinase (*YAK1*) resulted in fewer conidia and sclerotia production [77]. Penetration ability and H_2_O_2_ resistance were also compromised. MAPK Snf1 Activating Kinase (*SAK1*) yeast homologs are involved in the oxidative stress response via the high-osmolarity glycerol (HOG) pathway [132]. Interestingly, unlike other fungal SAK1 homologs [142], *B. cinerea* SAK1 does not appear to be significantly involved in light-dependent development, as *sak1* conidiophore and sclerotial formation were unaffected when grown in light and dark conditions respectively. *sak1* mutant also exhibited defective conidiation and loss of virulence. Virulence loss stems from loss of appressorium development and secondary metabolism of major phytotoxins. Therefore, the SAK1 cascade in *B. cinerea* seems to be a general regulator of metabolism and development rather than a specific stress response regulator.

The transmembrane osmosensors synthetic high osmolarity-sensitive protein 1 (SHO1) and synthetic lethal of N-end rule (SLN1) sense osmotic stress and serve as upstream biosensors of SAK1. Single and double mutants both exhibited growth, virulence, sporulation and sclerotium formation defects [105]. Interestingly, conidia size was significantly reduced in these mutants. *sln1* mutant failed to form any sclerotia but it produced a large amount of conidia regardless of lighting conditions. *sho1 sln1* double mutant grown in both light and dark exhibited reduced virulence and increased sclerotia production; however, sclerotia formed were much smaller. It was suggested SHO1 and SLN1 are involved in the light-dependent differentiation process. In particularly, SLN1 appears to be a major component in this process, as *sln1* failed to detect dark conditions and form sclerotia. However, SHO1 is also involved to a lesser degree, as *sho1 sln1* double mutant shifted from favoring sclerotia formation to conidiation. Future investigation into SHO1 and SLN1 interactions with light-dependent differentiation associated genes such as light-responsive transcription factors may reveal how SHO1 and SLN1 are involved in this process.

Nutrient accessibility is a key element for pathogen survival and host invasion. Conceivably, pathogenicity and growth of *B. cinerea* would logically decrease once nutrients are diminished. Phosphoenolpyruvate carboxykinase PCK1 allows for the generation of glucose in the absence of carbohydrates [133]. *pck1* deletion mutant displayed reduced conidiation, more but smaller sclerotia formation and delayed conidial and sclerotial germination. Also, the mutant conidia lost their ability of host penetration, which may partly explain its impaired pathogenicity. Besides, the mutant phenotype could be rescued by exogenous treatment of glucose. Thus, blocking of PCK1 activity may serve as a method to decrease *B. cinerea* spread and pathogenicity in agricultural crops.

Histone demethylation is a common way of regulating gene expression. Studies on dimethyladenosine transferase DIM5 and lysine(K)-specific histone demethylase KDM1 indicated their varying effects on pathogenicity and development, which will be discussed in later sections. Herein, deletion of one Histone 3 Lysine 4 (H3K4) demethylase JAR1 (JARID1) significantly suppresses conidiation, appressorium formation and virulence [58]. Curiously, sclerotia formation was promoted. A reduction in host-related stress adaptation and ROS production was thought to contribute to decreased virulence. Exogenous application of fructose rescued the appressorium defect, suggesting that loss of JAR1 leads to reduced uptake and utilization of fructose. Septin SEP4 plays an essential role in fungal development [82]. GFP-tagging of SEP4 revealed that absence of JAR1 prevented SEP4 from assembly within hyphae, suggesting JAR1 is required for proper SEP4 assembly and fungal tissue differentiation. Sclerotia formation may have increased to conserve energy because of absence of nutrient uptake and pathogenicity.

Synthesis of cell membrane constituents is crucial for cellular growth and development. Diacylglycerol O-acyl transferase 2 (DGAT2) catalyzes the final step of triacylglycerol (TAG) synthesis [143]. In fungi, TAG plays a significant role of maintaining lipid homeostasis and signal transduction. Deletion of *DGAT2* resulted in reduced sporulation and failure to form sclerotia due to lack of TAG [50]. As TAG comprises a high proportion of sclerotia membrane lipids, TAG shortage in the mutant is suspected to have limited sclerotia formation. Mutant virulence was also negatively impacted by low host penetration and reduced OA secretion. Hyphae formed from *dgat2* had swollen tips, preventing successful penetration of host tissue. The diminished activity of pH-dependent exogenous hydrolases was suspected to be diminished due to a lack of OA. Although TAG seems to affect OA levels in *B. cinerea*, it is unclear how *DGAT2* influences OA biosynthesis and secretion.

The alterative oxidative pathway serves as a substitute electron transport chain (ETC) pathway to bypass the typical cytochrome ETC pathway. It uses alternative oxidase (AOX) as the terminal oxidase and AOX was proposed to aid in regulating intracellular ROS [89,144]. Conidiation and lesion expansion were decreased in the *aox* mutants and the mutants produced a larger number of smaller sclerotia. As expected, *aox* mutants accumulated more ROS and exhibited greater oxidative stress sensitivity. Thus, the alterative oxidative pathway in *B. cinerea* significantly affects both development and pathogenicity.

The *B. cinerea* specific hyphal branching-related factor gene *HBF1* is a recent discovery whose function is still widely unknown. Mutant *hbf1* suffered from significantly altered conidia morphology, hyphal branching, sclerotium formation and virulence [88]. The virulence loss was suspected to be caused by reduction in appressorium numbers and delay of appressorium host penetration. *HBF1* expression was upregulated during growth and early invasion phase. The mutant phenotypes and transcriptomic data indicate that *HBF1* plays an essential role in early developmental and pathogenic stages. However, there are insufficient studies to determine the exact function of HBF1. Further research into the localization and functional domains of HBF1 would be necessary.

### 3.3. Signaling Events Leading to Conidiation

Conidia sporulation, which is the release of asexual spores from conidiophores, serves as the major inoculum for *B. cinerea* [8]. Conidiophores are developed from sclerotia and mycelia in early spring. Air currents caused by temperature fluctuations can lead to conidia release. Studies have indicated that conidia formation is light-dependent and light is necessary to stimulate sporulation. In this section, we will be mainly discussing genes affecting conidiation, growth and pathogenicity. As conidiation requires large amounts of nutrients acquired from host tissue invasion, nutrient availability is inheritably linked with fungal growth [145,146]. It is therefore not surprising that genes affecting conidiation would also affect growth, which likely indirectly affects virulence.

The cAMP signaling pathway is essential for environmental signal perception and transduction. It is proved to be involved in pathogenesis and differentiation in many fungal species. In this process, cellular cAMP level has to be regulated to ensure normal signaling. At the very beginning, cAMP synthesis from ATP is catalyzed by botrytis adenylate cyclase (BAC). When *BAC* was knocked out, the mutant exhibited reduced vegetative growth and lesion expansion [130]. Interestingly, leaves inoculated with *bac* mutants failed to develop any conidia, suggesting that BAC is essential for sporulation. However, low cAMP levels were still detected in *bac* mutants, indicating that other similar enzymes such as guanylate cyclase might be able to partly compensate for the loss in BAC function.

Small GTPases are GTP-binding hydrolases enzymes which regulate a wide range of cellular process based on GTPase phosphorylation state. Deletion of RAS-GTPase *RAS2* resulted in a delayed conidia germination, pathogenicity phase as well as reduced growth [81]. Interestingly, exogenous application of cAMP rescued the mutant to WT phenotype, suggesting that RAS2 is connected to the cAMP pathway probably by activating BAC.

MAP kinase cascades are conserved signaling modules for eukaryotic biology. The botrytis osmosensor BOS5 and OS4 are upstream histidine kinase components of the high osmolarity glycerol (HOG) pathway that significantly influences development, virulence and stress resistance in *B. cinerea*. Deletion of the MAPKK and MAPKKK-encoding genes *BOS5* [113] *OS4* [118] in *B. cinerea* resulted in significantly impaired hyphal growth and increased sensitivity to fungicides and ionic stress. Both mutants were unable to form any conidia or infect plant leaves. Furthermore, cucumbers inoculated with *bos5* and *os4* spores only produced sparse mycelium fibers without conidiation. The downstream HOG pathway component SAK1 exhibited reduced phosphorylation in *bos5* and *os4* mutants, supporting roles of BOS5 and OS4 in proper HOG pathway signaling.

Transcriptional regulators are essential components of signal transduction to control gene expression. The response regulator RRG1 is the putative regulator of HOG pathway in *B. cinerea* [101]. *rrg1* deletion mutant failed to form conidia and had increased fungicide and ionic stress sensitivity but their virulence was not affected. The decreased virulence in mutants of upstream HOG pathway components such as BOS5 and SAK1 suggests that there are other regulators controlling pathogenicity. Regulator 1 (REG1) belongs to a novel class of fungal transcription regulators involved in pathogenicity and morphology [147]. The knockout mutant exhibited reduced expression of oxidative stress response genes and virulence. *reg1* mycelium could penetrate host tissue but was unable to produce lesions. It is suspected that the inability to cause lesions is due to the loss of phytotoxin production. Analysis of *sak1* mutant revealed that *REG1* expression is dependent on SAK1 and might have contributed to the virulence loss in *sak1*. Suppressor of Kre Null 7 (SKN7) is another downstream transcription factor in the HOG pathway [69]. SKN7 homologs are associated with fungal development and various stress adaption. In consistency, deletion of *SKN7* in *B. cinerea* resulted in fewer sclerotia, no conidia and increased sensitivity to H_2_O_2_.

Light-responsive transcription factor 1 (LTF1) has a significant role in regulating gene expression against the detrimental effects of light [129]. Although deletion of *LTF1* decreased virulence and growth, it caused excessive but precocious conidiation. Virulence and growth could be restored by antioxidant application, indicating that the virulence and growth loss could be due to ROS accumulation. On the other hand, overexpression of *LTF1* caused *B. cinerea* to produce more aerial mycelia and less conidia. Similarly, deletion of *LTF3* promoted conidiation under all light conditions but no mature conidia could be produced [78]. *ltf3* conidiophores would develop hyphae and branch off into secondary conidiophores, producing a mycelia-like structure. Conversely, overexpression of *LTF3* resulted in significant suppression of conidiation development. These results indicate that both LTF1 and LTF3 are general repressors of conidia development and LTF1 suppresses it through regulating ROS homeostasis while LTF3 does so via unknown means. It is likely the two LTFs are functionally overlapping, as the double mutant *ltf1 ltf3* exhibited additive mutant phenotypes [78].

Homeobox transcription factor genes (*HOX*) encode highly conserved master developmental regulators known to play major roles in fungal growth and differentiation. The *B. cinerea* genome contains 9 *HOX* genes. So far, only *HOX8* has been studied in detail [106]. Mutants of *HOX8* exhibited slow vegetative growth, reduced infection efficiency and disease progression. As *HOX8* is expressed at higher level in conidia, it is not surprising that conidiogenesis was strongly affected by *HOX8* deletion and *hox8* mutants produced a small number of deformed conidia. It is suspected that misshaped conidia had contributed to the loss of infectivity. The *hox8* phenotypes are quite unique among known *B. cinerea* mutants, indicating HOX8 may be part of an unknown signaling pathway.

The pentose phosphate pathway (PPP) is a highly conserved metabolic pathway playing a major role in NADPH synthesis. The initial reaction is catalyzed by the enzyme 6-phosphogluconate dehydrogenase (PGD). First discovered as a downregulated protein in *noxR* mutants, PGD appeared to be a major development and virulence factor regulated by the NOX complex. *pgd* mutants exhibited impaired virulence, growth and sporulation [95]. It is unclear how sporulation and virulence are affected but reduction in NADPH synthesis is believed to influence pathogenicity through affecting conidiophore and appressorium formation.

Hexose kinases play key roles in sugar catabolism by phosphorylating glucose and fructose. Hexokinase *hxk1* mutants displayed severe growth retardation, impaired lesion formation and conidiation [110]. *hxk1* conidia exhibited failed germ tubes elongation, significantly affecting the invasion ability of the mutants. The loss of sugar phosphorylation likely disrupted sugar metabolism, leading to *hxk1* mutant phenotypes. On the contrary, deletion of glucokinase *GLK1* had no detectable phenotypic difference from WT [110]. As enzymatic activity analysis indicated that GLK1 and HXK1 both significantly contribute to phosphorylating glucose, GLK1 likely has other redundant enzymes in *B. cinerea*.

### 3.4. Infection and Pathogenicity Mechanisms

*B. cinerea* mostly initiates infection through conidia spores landing on host plants. They germinate and form appressoria used for host tissue penetration [8]. Evading host immunity and production of molecules leading to host cell death are key invasion strategies in *B. cinerea*. Genes discussed in this section are highly specific and only affect discrete parts of the pathogenic process. We will first focus on genes essential during the pre-penetration stage of *B. cinerea*, where germinating conidia must adhere to plant surfaces and avoid elimination by host defenses.

Conidia of *B. cinerea* initially passively attach to plant surfaces via hydrophobic forces [148]. In later germination stages, an adhesive extracellular matrix (ECM) composed of carbohydrates and proteins is secreted. Secretomic analysis identified β-glucosidase Sad1 Unc-84 Domain Protein (SUN1) as a major component of *B. cinerea* adhesive matrix [149,150]. SUN protein family members are ascomycete specific and classified by a cysteine-rich SUN domain. Disruption of *SUN1* significantly decreased conidia and mycelia adhesion to host surfaces and interfered with reproductive structure formation mostly through decreased ECM [92]. The reduced adherence contributed to the lower capacity of initiating a successful infection of *sun1* mutants, thus exhibiting largely reduced symptoms in the host plants. Interestingly, sclerotia formed by *sun1* mutants were significantly increased while conidiophore formation was halved, suggesting the contributions of β-glucosidase to fungal development as well. In addition to the reduced ECM adhesion, the change in conidiation significantly contributed to reduced dispersal of conidia.

Host cell wall and antifungal SM secretions are key components of plant immunity against microbial pathogens. *B. cinerea* needs to evade plant defenses for successful invasion. Apoptosis-like programmed cell death (PCD) in fungi can be induced by plant-secreted antifungal SMs. Thus, deactivation of antifungal SMs is necessary to prevent apoptosis. In *B. cinerea*, antifungal SMs are inhibited by baculovirus inhibitor of apoptosis protein (IAP) repeat BIR1 and its regulator protein nuclear mediator of apoptosis (NMA) [97]. *BIR1* overexpression led to increased lesion expansion and reduced apoptosis markers. On the other hand, *bir1* knockdown mutant showed decreased virulence and increased PCD. Further fluorescence imaging showed that *BIR1* was strongly expressed during the first 24 h post-inoculation, indicating that BIR1 has a central anti-apoptosis role in *B. cinerea* during initial infection stages. *nma* deletion mutants exhibited higher hyphal growth and reduced apoptosis, while *NMA* overexpression led to increased PCD with no change in growth rate. However, both *NMA* deletion and overexpression strains caused slightly milder symptoms in the host. These indicate that NMA has a pro-apoptosis function in *B. cinerea* but it does not act as a major regulator of apoptosis and has minor effects on pathogenicity.

Intracellular accumulation of toxic compounds such as phytoanticipins and phytoalexins can severely impact pathogen survival on host plants. Active efflux of fungitoxic compounds allows fungi to gain resistance against SMs, antibiotics and fungicides. However, active efflux transporters are not well studied in plant pathogens. The Major Facilitator Superfamily transporter, MFSG plays an essential role in *B. cinerea* resistance to isothiocyanates (ITCs) [87]. *MFSG* deletion resulted in a significant virulence reduction and decreased survivability *in planta*, indicating that MFSG is a key component of resistance against antifungal compounds in *B. cinerea*. *mfsg* mutants exhibited varying degrees of sensitivity to products of phytoalexin glucosinolate hydrolysis, suggesting that MFSG may also export other antifungal compounds, the identities of which are unknown.

Plants also utilize ROS secretion to disrupt cellular processes in pathogens for defense. Redox systems have a central role in maintaining oxidative homeostasis in fungi. Two major redox systems in *B. cinerea* are the thioredoxin and glutathione (GSH) systems. Although thioredoxin and GSH systems participate in both enzymatic and antioxidative processes, we will be mainly focusing on their antioxidative properties. Thioredoxin system includes two components, thioredoxins and thioredoxin reductase. Two thioredoxins encoding genes TRX1 and TRX2 and one thioredoxin reductase encoding gene TRR1 were identified [53]. As expected, knockout mutants *trx1* and *trr1* exhibited increased oxidative stress sensitivity and impaired virulence. *trr1* also showed enhanced H_2_O_2_ accumulation, in which oxidative stress from excess ROS caused decreased vegetative growth rate. In contrast, *trx2* had no phenotypic changes, indicating a major role of TRX1. Deletion of glutathione reductase gene *GLR1* only resulted in slightly decreased virulence and appressorium formation, while *glr2* did not have any phenotypic changes [53]. This indicates that the GSH is not a major redox system in *B. cinerea.* Conceivably, studies on enzymatic activity revealed overlapping functions between the thioredoxin and GSH systems [53], whereby thioredoxin contributes to the majority of antioxidative processes in *B. cinerea* while GSH is a redundant and minor redox system.

CWDEs are essential in *B. cinerea* as they assist penetration through the plant cell walls and the breakdown of host tissue post-infection. Pectin methylesterase 1 (PME1) is secreted early in the invasion phase to hydrolyze pectin, a major plant cell wall component [102]. *pme1* deletion mutant exhibited a four-fold reduction in PME activity. Consequently, mutant pathogenicity was considerably decreased. PME activity was not totally eradicated in *pme1*, indicating the presence of other PMEs. Investigating these PMEs is essential for fully understanding the mechanisms of PMEs in *B. cinerea* virulence.

The sugar side chains of pectin arabinan are depolymerized by endo-arabinasanse ARA1. ARA1 seems to be the sole arabinasanse in *B. cinerea*, as indicated by knockout mutants *ara1* being deficient in arabinan-degrading activity [65]. *ara1* exhibited decreased growth on medium with arabinan as the only carbon sources and severely decreased lesion expansion on only *A. thaliana*. No virulence alteration was observed in other tested hosts including *Nicotiana benthamiana* and tomato plants. It was proposed that the high arabinan content in *A. thaliana* serves as physical barriers to impede the spread of *ara1*. This suggests that ARA1 only plays a major pathogenicity role in plants with cell walls rich in arabinan. In comparison, deletion of the endopolygalacturonase *PG2* gene resulted in various degrees of host-dependent delay in lesion expansion rate [128]. It is suggested that difference in host cell wall composition might be the cause of variation in *pg2* virulence in different plants.

Endo-β-1,4-xylanases (XYNs) digest xylan, the hemicellulose component of the plant cell wall. Most fungal xylanases belong to two major families, glycosyl hydrolase families 10 (GH10) and 11 (GH11) [136]. XYN10A and XYN10B belong to the GH10 family while XYN11A, XYN11B and XYN11C belong to the GH11 family. Disrupting *XYN11A* resulted in 30% decrease in endoxylanase activity and severe virulence defects in lesion formation and expansion [151]. Strains with simultaneous silencing of all five *XYN* genes exhibited slight reduction in growth rates and lesion growth [136]. XYN activity in the RNAi mutant was reduced to the same level as *xyn11A* mutants. Expression analysis of the RNAi mutant indicated that *XYN11A* and *XYN11C* genes were overexpressed. This could be the result of homologous gene compensation. Further studies involving deletion of individual *XYN* genes would be necessary to understand the function of *XYN* gene family in *B. cinerea*. XYL1 is another xylanase in *B. cinerea* belonging to the SGNH (Ser-Gly-Asn-His motif) hydrolase subfamily. Deletion *xyl1* mutants exhibited significant reduction in virulence [107]. Interestingly, XYL1 protein with impaired xylanase function was still able to induce plant cell death, suggesting that the cell death induced by XYL1 is separated from its xylanase activity. Further experiments on plant immunity indicated that XYL1 induces plant cell death by acting as pathogen-associated molecular patterns (PAMP) and conferring PAMP-triggered responses (PTI) in host plants.

The lipase gene (*LIP1*) encodes a lipolytic serine esterase, which is commonly known as lipase. Lipases are secreted by pathogenic fungi and used to degrade the outer wax layer of the cuticle [112]. As expected, deletion of *LIP1* resulted in significant loss of lipase activity. However, *lip1* retained full WT-like virulence. Although not all lipase secretion was eliminated in *lip1*, it appears that lipases do not play major pathogenic roles in *B. cinerea*.

*B. cinerea* utilizes the D-galacturonic acid (DGA) catabolic pathway to convert DGA, a monosaccharide abundant in pectin polysaccharide cell walls into pyruvate and L-glyceraldehyde. The pathway involves two nonhomologous galacturonate reductase GAR1 and GAR2, L-galactonate dehydratase LGD1 and 2-keto-3-deoxy-L-galactonate aldolase LGA1 [57]. All single knockout mutants exhibited strongly reduced virulence on DGA rich plants such as *A. thaliana* and *N. benthamiana*. A WT-like pathogenicity was seen on plants with low DGA content like tomato. The DGA catabolic intermediates appeared to inhibit *B. cinerea* growth, as mutants with disrupted DGA catabolic pathways exhibited reduced growth rates when incubated with DGA catabolic intermediates. It is unclear if the DGA catabolic pathway contributes to virulence as a CWDE or aids in fungal growth as an alternative nutrient source.

Monocarboxylate transporter MCTA is a major pyruvate importer in *B. cinerea* and is essential to access nutrients from degraded plant cell walls [119]. Deletion of *MCTA* resulted in decreased sclerotia formation, virulence and growth on acetate or pyruvate carbon media. However, conidiation did increase two-folds. Currently, further investigations are needed to explain the role of MCTA in the virulence of *B. cinerea*. Research on the fungal pH-responsive transcription factor PacC [116] suggested that *B. cinerea* necrosis activity is induced by low cellular pH levels, which can occur by importing acidic compounds such as pyruvate into fungal cells.

Conidial germination is essential for *B. cinerea* colonization. Germ tubes grown from germinating conidia develop into appressoria upon maturation. Chitin, an essential structural component of fungal cell walls, is synthesized by chitin synthases (CHSs). The functions of CHS4, CHS6 and CHS7 were previously investigated [51]. Disruption of *CHS6* resulted in significant decrease in hyphal growth, conidiation and germination in heterokaryotic strains. A strong virulence reduction was also observed. No *chs6* homokaryotic strains were isolated, indicating that CHS6 may be an essential enzyme for *B. cinerea*. *chs4* and *chs7* mutants had no changes in CHS activity, indicating their redundancy in chitin biosynthesis. Interestingly, the pathogenicity of *chs7* mutants only decreased in *A. thaliana* among all the tested plant species, suggesting that CHS7 may act as a host-specific virulence factor.

Cytoskeleton is a dynamic protein network composed of microtubules, microfilaments and intermediate filaments. Microfilaments consists of polymerized actin subunits. In filamentous fungi, actin (ACT) is essential for hyphal growth and enzyme secretion. Deletion of a conserved actin gene *ACTA* in *B. cinerea* resulted in reduced growth, sporulation and virulence [135]. Proteomic analysis of *actA* revealed a decreased secretion of 11 CWDEs including the aforementioned XYNs. Actin is essential for normal development and pathogenicity in *B. cinerea*, especially for virulence factor secretion. Kinesins are microtubules-based motor proteins crucial for intracellular transportation and cell division. The kinesin-like protein KLP7 belongs to a novel kinesin subfamily exclusively found in fungi [76]. As kinesins are linked with fungal growth and development, it is not surprising that disruption of *KLP7* resulted in observable defects in hyphal growth, sporulation and conidium germination. The hyphae of *klp7* were irregular and stunted, which may lower the penetration ability of the mutant. Analysis of the known virulence factors indicated severely decreased secretion of OA, polygalacturonase (PG) and PMEs. Thus, KLP7 may be involved in transportation of secretory vesicles containing these virulence factors.

Identifying growth direction is essential for proper hyphal growth, reproductive structure formation and especially in host tissue penetration for fungal pathogens. Deletion mutants of orthologous components in the yeast polarity complex were generated, including the scaffold proteins bud emergence protein 1 (BEM1) and factor arrest protein FAR1, the guanine nucleotide exchange factor cell division control protein 24 (CDC24) and septin SEP1 [84]. BEM1 is a major scaffold protein in the polarity complex and the deletion mutants of *bem1* exhibited significantly reduced conidiation and sclerotia formation. The reduction in colonization and virulence of *bem1* could be because appressoria structure and germination rates were disrupted in the mutants. In other fungi, CDC24 is known to be critical for fungal growth. Heterokaryotic *cdc24* strains of *B. cinerea* were used in this study as no homokaryotic strains could be isolated. The *cdc24* mutants mainly exhibited growth retardation, conidium deformation and host penetration defects. Most conidia from *bem1* and *cdc24* mutants could not germinate and the few germinated ones gradually stopped growing. These results indicate that BEM1 and CDC24 significantly contribute to conidium germination, hyphal growth and host penetration. SEP1 plays a role in formation of septum, cell wall structures essential for fungal cell division. Deletion of SEP1 resulted in decreased growth and inability to form reproductive structures such as conidiophores and sclerotia. The reduced virulence was also observed; however, it is unclear of the reasons for pathogenicity loss. *far1* mutant had no observable phenotypic changes, suggesting it plays a minor role in the *B. cinerea* polarity complex. Determining specific interactions between CDC24 and SEP1 with BEM1 may yield further knowledge about major signaling pathways in *B. cinerea*, as BEM1 significantly affects multiple cellular processes.

Following hyphal growth, the germ tube tips become swollen and form compound appressoria for host tissue penetration. Fungal appressorium formation requires the transmembrane protein tetraspanin. In ascomycetes, the punchless 1 (PLS1) tetraspanin protein is commonly associated with virulence [111]. Deletion of *PLS1* resulted in female sterility and reduced virulence in *B. cinerea*. Interestingly, appressoria-like swollen hyphal tips were still observed in *pls1* deletion mutants but they were unable to penetrate host tissue. This suggests that PLS1 may indirectly initiate penetration by interactions with proteins associated with appressorium development such as BHP1, BHP2, JAR2 or HBF1.

After penetration, *B. cinerea* secretes numerous virulence factors to induce host cell damage and necrosis. Generally, these factors include ROS and fungal toxins. The two H_2_O_2_ generating enzymes in *B. cinerea* are Cu-Zn-superoxide dismutase SOD1 and glucose oxidase GOD1 [75]. *sod1* deletion mutants exhibited significantly retarded lesion development, indicating that *SOD1* is a key gene in H_2_O_2_ generation. However, *god1* mutants had no observable phenotypic change and *GOD1* expression level in WT was very low. Thus, GOD1 may be a minor H_2_O_2_ generation enzyme in *B. cinerea*.

The fungal glucose methanol choline (GMC) oxidoreductase conidial germination-associated factor 1 (CGF1) is a novel virulence factor associated with extracellular H_2_O_2_ production in basidiomycetes. *B. cinerea cgf1* deletion mutants exhibited impaired conidia production and germination and drastically reduced formation of invasive germ tubes and appressia [134]. Further analysis revealed that CGF1 mediates virulence by promoting endogenous ROS production, which in turn simulates infection processes such as appressorium formation. Interestingly, secretion signals in CGF1 were found via bioinformatics analysis, suggesting that CGF1 may be a secreted protein and have extracellular functions. In order to further understand functions of CGF1, more studies focusing on identifying subcellular localization and functional domains of CGF1 are necessary.

The phytotoxin botrydial is secreted only by *B. cinerea*, which induces necrotic lesions in hosts. The biosynthesis pathway of botrydial is still unknown. So far, only the P450 monooxygenase BOT1 has been identified as part of this synthesis pathway [125]. As expected, *bot1* deletion mutants no longer produced botrydial and were significantly less pathogenic. Interestingly, *BOT1* expression appeared to be regulated by a signaling cascade including BCG1 (detailed in earlier section) and suppression of *BOT1* may be a major cause of virulence loss in *bcg1*.

Regulator of IME2 (RIM15) is a Per-Arnt-Sim (PAS) kinase downstream of Mitogen-activated protein Kinase-Kinase (MKK1) and the cell wall integrity (CWI) pathway [52]. *RIM15* deletion mutants resulted in slight decreased growth rates but significantly increased OA production and extracellular protease activity. The significantly increased enzymatic activity is likely due to the secreted hydrolases being optimal under acidic conditions with extra OA. Interestingly, virulence of *rim15* was not significantly different from WT, suggesting that acidification of host tissue is not a major contributor to *B. cinerea* pathogenicity as in other fungi.

Fungal pathogens exploit plant defense systems by inducing a HR in host plants. In doing so, a rapid necrosis of plant tissue surrounding infected areas would occur. The snod-prot-like protein SPL1, a member of the cerato-platanin family, is one of the most abundant proteins in the *B. cinerea* secretome [71]. Knockout *spl1* mutants exhibited weaker HR and decreased lesion growth. Expression of *B. cinerea SPL1* gene into *Pichia pastoris* yeast cells resulted in purified protein able to trigger HR in infected tissue, indicating SPL1 is a major fungal phytotoxin.

Proteins with CFEM domains (common in several fungal-specific eight-Cys-containing domain of extracellular membrane proteins) are fungal-specific extracellular membrane proteins associated with virulence in pathogenic fungi [152]. In *B. cinerea*, CFEM1 induces chlorosis in host leaves [115]. Deletion of *CFEM1* significantly decreased pathogenicity, conidiation and increased sensitivity to osmotic and oxidative stresses. In addition to lack of chlorosis induction, the increased sensitivity to H_2_O_2_ may have contributed to virulence loss in the *cfem1* mutant. Currently, the exact epitopes of CFEM1 that triggers the chlorosis pathway in host plants are unknown. Contrary to the initial assumption of being membrane-bound by a glycophosphatidylinositol (GPI) anchor, secretion signal peptide was found in CFEM1, suggesting it may be a secreted protein. Discovering the epitopes of CFEM1 and figuring out whether CFEM1 is membrane-bound or secreted would be critical to understanding the pathogenic mechanisms of CFEM1.

In filamentous fungi, the target of rapamycin (TOR) pathway links nitrogen sensing to cellular growth and development. It can be suppressed with rapamycin [120]. The FK506-binding protein FKBP12 is required for normal rapamycin disruption of the TOR pathway. As expected, deletion of *FKBP12* resulted in rapamycin-resistant mutant strains, indicating that *B. cinerea* FKBP12 is functionally orthologous to other fungal FKBP12s in function. Interestingly, *fkbp12* deletion had varying effects on virulence in different *B. cinerea* strains. In strain B05.10, pathogenicity was unaffected on tomato and grape but exhibited improved colonization rates on apple and cucumber fruits. However, in strain T4, *fkbp12* exhibited reduced lesion expansion on tomato and bean leaves. It was suggested that FKBP12 inactivation could affect virulence factor secretion such as CDWEs and phytotoxins to various degrees in different strains.

### 3.5. Generalist Genes–Mutants with Defects in Multiple Aspects of B. Cinerea Biology, Including Growth, Virulence and Sclerotia or Conidia Development

In this section, we will be discussing master contributors to *B. cinerea* biology. These transcription factors and components of signaling systems regulate a broad range of downstream factors. Thus, it is not surprising that their disruptions would yield multiple developmental and virulence defects.

The Ca^2+^/CN signaling pathway is a critical fungal system required for normal virulence, hyphal growth and morphology in pathogenic fungi. CN belongs to the serine/threonine-protein phosphatase 2B (PP2B) family. Deletion of the catalytic subunit of calcineurin (CNA) in *B. cinerea* resulted in avirulent mutants with significantly reduced growth rates and inability to form conidia [59]. Few sclerotia were produced only after long incubation in dark conditions. Interestingly, the color of *cna* colonies changed over weeks from dark to light-colored, suggesting a loss of melanization. However, it remains unclear whether the change in melanization significantly contributed to other mutant phenotypes such as virulence. Furthermore, CN activity is regulated by regulators of CN (RCN1). Knockout mutant *rcn1* exhibited retarded hyphal growth similar to *cna* and WT strains treated with CN inhibitors, indicating RCN1 functions as a positive growth regulator [59]. This was further supported by the fact that mutations in RCN1 phosphorylation sites resulted in deactivation of CN-promoting activity.

Cyclophilins are ubiquitous in all organisms and they are known for forming complexes with the drug cyclosporin A (CsA) to inhibit CN function. Like CN, cyclophilins are also cellular components involved in fungal morphogenesis and pathogenesis but their mechanism is unclear. In *B. cinerea*, cyclophilin A is encoded by *BCP1* and deletion of the gene resulted in decelerated lesion expansion in the host [85]. In the study, three cyclopilian-dependant genes (CPD) were identified by cDNA macroarray analysis, suggesting BCP1 plays a *B. cinerea* specific role in promoting *CPD* expression. Interestingly, CN inhibition with CsA resulted in different phenotypes, including abnormal hyphal morphology and failure to form appressoria. These differences indicate that CN and cyclophilin A affect distinct aspects of morphogenesis and virulence in *B. cinerea.*

The CN-Responsive Zinc Finger 1 (CRZ1) is a transcription factor downstream of CN in *B. cinerea*. Deletion *crz1* mutants showed altered hyphal morphology and a strong reduction in growth, conidiation and sclerotia formation [55]. *crz1* also displayed increased sensitivity to oxidative (H_2_O_2_) and osmotic (Li^+^ and Ca^2+^) stress. *crz1* penetration ability was severely impaired, indicated by delayed necrosis symptoms on intact leaves and WT-like pathogenicity on pre-wounded leaves. Treatment of *crz1* conidia with CN inhibitors resulted in further additive growth defects to *crz1*, indicating that *crz1* is not the sole target of CN. Gene expression analysis revealed that the expression of most of CRZ1-dependent genes were downregulated by *CRZ1* deletion, indicating that CRZ1 acts as a transcriptional activator. However, it is uncertain whether CRZ1 directly regulates expression of these genes. Functional study of these genes is paramount to determine the effects of CRZ1 on global gene expression regulation.

PP2A is a serine/threonine protein phosphatase 2A essential for fungal growth and pathogenicity and it plays similarly critical roles as CN in signaling pathways including fungal growth, differentiation and metabolism. *B. cinerea* heterokaryotic *PP2A* deletion and *PP2A* silenced strains were generated, as homokaryotic *PP2A* deletion strains could not be isolated [123]. *pp2A* mutants exhibited severely defective growth and complete failure in sclerotia formation and virulence. The growth rate of *PP2A* deletion and silenced strains under oxidative stress was further decreased, indicating an increased sensitivity to H_2_O_2_-induced oxidative stress in the mutants. Similar change in oxidative stress sensitivity to H_2_O_2_ was observed in mutants of redox system genes such as *noxB, noxR, trx1* and *trr1*, suggesting that PP2A may act in conjugation with these pathways to regulate ROS production.

After activation of the cAMP signaling pathway, the cAMP cascade recovers by hydrolyzation of cAMP to AMP via phosphodiesterases (PDEs). Two PDEs, a low-affinity PDE1 and a high-affinity PDE2, have been described in filamentous fungi. Deletion mutant *pde1* exhibited WT-like phenotypes, while *pde2* displayed severe growth, sclerotium formation defects and decreased invasive growth [99]. *pde2* was also unable to sense light and surfaces for conidiation and germination. Intriguingly, intracellular cAMP levels in the *pde2* were slightly lower compared to WT. It is proposed that unknown linkages between PDE2, BAC and PKA1 may be the cause of such anomaly, as BAC and PKA1 regulates cAMP levels and they had altered activity in *pde2*. Overexpression of *PDE1* in *pde2* rescued the WT-like phenotype. Therefore, PDE2 is a major PDE in *B. cinerea*, while PDE1 is a minor PDE with redundant functions.

Signaling components are connected by small GTPases, a group of hydrolase enzymes which transform from inactive to active state by binding to GTP. Small GTPases are classified by the conserved G domain, where GTP binding occurs. The Ras small GTPase family (first discovered in rat sarcoma) is the most studied and is known for being involved in fungal development and pathogenicity. Deletion of *RAS1* gene in *B. cinerea* resulted in stunted hyphal growth, avirulence and inability to produce conidia [47]. Phosphorylation assays indicated that SAK1 was not phosphorylated in *ras1*, suggesting that RAS1 may act upstream of the SAK1 pathway.

Another small GTPase family is the RAS homologous (Rho) family, known to plays roles in cytoskeleton development and cell cycle. A prominent Rho signaling cascade is composed of the upstream RAS-related C3 (RAC), the downstream p21-activated kinase CLn activity dependent (CLA4) and effector cell division control protein 42 (CDC42). As expected, single deletion mutants *rac*, *cla4* and *cdc42* all exhibited the same phenotypes of decreased hyphal growth, defective sclerotium formation and inability to invade host tissue [47,73,126]. In both *rac* and *cla4*, the cell cycle was affected as mutants were unable to undergo mitosis to form conidia. Interestingly, that *cdc42* mutants did produce misshaped conidia but with germination defects, suggesting CDC42 plays essential roles in *B. cinerea* growth, virulence and germination but not in conidia formation. Analysis with phosphorylation assays suggested that CLA4 may be linked to cyclin-dependent kinase CDK1, a regulator of mitosis entry, as phosphorylated CDK1 levels dropped significantly in *cla4.* It is suspected that CLA4 is linked to CDK1 by the signaling component WEE1 in a similar manner to other fungi [153]. Future analysis of WEE1 would be necessary to confirm its role as a linkage protein between CLA4 and CDK1.

RHO3 is another member of the Rho GTPase family and it plays varying roles in cell growth, division and protein secretion in filamentous fungi by influencing actin localization. *rho3* deletion mutant showed decreased growth, conidiation, appressorium formation and virulence but enhanced sclerotium formation [91]. Besides, more ROS generation was observed in the mutant hyphal tips. Mitochondria staining experiments revealed an absence of mitochondria distribution in hyphal tips of *rho3*. As a result, not only did the loss of hyphal mitochondria accumulation affect germ tube and appressorium development due to reduced energy production but also virulence by attenuating intercellular ROS.

The Rab (Ras-related in brain) is a small GTPase family involved in the secretory pathway mechanisms of vesicle docking and fusion. Disruption of the Rab GTPase SAS1 (something about silencing) resulted in decreased growth, conidiation and lesion expansion in addition to failed sclerotium formation and abnormal hyphal morphology [46]. Analysis of CWDE activity indicated secretion of PGs and XYNs was significantly suppressed while gene expression was not affected. The mutants were able to penetrate and establish in host tissue but they failed to cause further symptoms. The reduction in fungal growth and development was suspected to be due to the central role SAS1 plays in vesicle secretion containing synthases and hydrolases, essential enzymes for cell wall growth, to the fungal cell wall. In summary, the small GTPase SAS1 contributes to both *B. cinerea* development and pathogenicity by its involvement in enzymes secretion.

The *MTG2* gene encodes mitochondrial GTPase 2, a member of the Obg GTPase family (SpoOB-associated GTP-binding proteins), which is critical for mitochondrial ribosome assembly in yeast. Deletion of *MTG2* gene resulted in slower growth rates and reduced conidium formation, germination and sclerotium formation in *B. cinerea* [30]. Interestingly, *mtg2* mutant also displayed increased sensitivity to cell wall disruption, oxidative and osmotic stress. The loss of mitochondrial ribosomes in *mtg2* may have caused the disruption of development, pathogenicity and environmental stress intolerances.

Transcription factors regulate gene expression by promoting or suppressing RNA polymerase II binding to DNA. So far, only a few transcription factors have been studied in *B. cinerea*. The pH-responsive transcription factor PACC, the terminal component of the Pal/Pac pathway, is of particular high interest, as it is essential for the necrotrophic activities of *B. cinerea* [116]. The fungal-specific Pal/Pac pathway regulates pH-dependent gene expression, ensuring that extracellular enzymes are only produced at suitable environmental pH. *pacC* deletion mutant inoculated on host tissues with neutral pH (bean and cucumbers) exhibited significant reduction in growth and virulence with no conidiophore or sclerotium formation. Secretomic analysis of *pacC* revealed a decrease in OA, ROS and CWDE secretion, suggesting the change in virulence is due to reduction in virulence factors secretion. Interestingly, *pacC* mutants grown on host tissue with acidic pH such as fruits of apple (pH 4.0), cherry tomato (pH 4.7) and grape (pH 3.8) plants exhibited a WT-like phenotype, demonstrating the importance of initial environmental acidification to induce the onset of *B. cinerea* pathogenicity. This is unlike *rim15* mutants, where post-infection acidification is unnecessary for virulence of *B. cinerea*.

The ubiquitous and highly conserved MADS-Box transcription factors have been demonstrated to be essential for fungal development and virulence but the regulatory mechanisms of MADS are not well understood. Deletion of the myocyte enhancer factor (MEF2)-type *MADS1* gene in *B. cinerea* resulted in a mutant with slower vegetative and lesion growth and produced larger conidia [68]. Interestingly, conidiation instead of sclerotia formation was induced in dark conditions, indicating *MADS1* deletion disrupted the expression of light-responsive genes. It was suggested the loss of MADS1 caused an increase in photoreceptor-encoding gene expression and therefore light sensitivity. Analysis of *mads1* proteome revealed the secretion proteins SEC14 and SEC31 as potential targets of MADS1. *sec14* and *sec31* deletion mutants both exhibited reduced lesion expansion and extracellular protein secretion, indicating they are essential for *B. cinerea* pathogenicity [68]. Studies on subcellular localization and secretomics of *sec14* and *sec31* would be necessary to understand the full roles of these secretory proteins.

The subunit of the Spt-Ada-Gcn5-acetyltransferase (SAGA) transcriptional regulator Suppressor of Ty (SPT3) was previously found to regulate growth and virulence in *Candida albicans* and *F. oxysporum* [154,155]. In addition to losing conidiation, sclerotia formation and some pathogenic properties, *SPT3* knockout mutants exhibited increased sensitivity to fungicides, oxidative stress and cell wall disrupting compounds [123]. Interestingly, *spt3* mutants grown without light showed accelerated growth rates, suggesting SPT3 may regulate expression of some light-dependent growth genes. In sum, the SAGA transcriptional regulator in *B. cinerea* regulates development, virulence and stress-induced gene expression.

Gene expression can be controlled by alternation of chromatin structure and DNA accessibility with histone modifications. Among varied modification, histone methylation plays pivotal roles by attaching methyl groups to lysine and arginine residues of histone 3 and 4, allowing for transcriptional activation or repression [96]. Deletion of histone 3 lysine 36 (H3K36) demethylase *KDM1* gene caused *B. cinerea* to produce sclerotia in both light and dark conditions and affected excessive light and oxidative stress tolerance. As expected, growth was only impaired under stress conditions. KDM1 is critical for *B. cinerea* pathogenicity, as indicated by its mutant mycelia unable to form effective penetration structures. Thus, KDM1 is required for light-dependent development, stress resistance and formation of penetration structures (compound appressoria). Interestingly, insertion of *KDM1* orthologs from other ascomycetes into *kdm1* did not restore WT phenotype, indicating that the KDM1 functions may be species-specific. Histone H3 lysine 9 (H3K9) methyltransferase DIM5 (defective in methylation) deletion led to impaired hyphal growth, pathogenicity and reduced conidiophores and sclerotia production [117]. Decreased virulence may be the result of changes in expression of virulence factors. Indeed, expression assays revealed downregulation of virulence-related genes including *SOD1*, *SPL1*, VELVET complex genes and *BOT* genes. However, the cause of compromised development in *dim5* is unknown. Since histone methylation by DIM5 regulates significant amounts of virulence factors, further investigations into *DIM5* gene expression profile may yield novel virulence genes targeted by DIM5.

DNA methylation in *B. cinerea* is achieved by a complex containing heterochromatin protein HP1 and DNA methyltransferase DIM2. Single deletion mutants *hp1* and *dim2* did not have any noticeable defects, indicating DNA methylation may not play major virulence roles [117]. Further research may be necessary to understand the function of HP1 and DIM2 in *B. cinerea*.

The nucleolar protein NOP53 plays important roles in ribosome assembly and pre-rRNA processing in yeast. *B. cinerea nop53* deletion mutant exhibited enhanced stress sensitivity, impaired virulence and defective conidia, sclerotia and infectious structures formation [122]. The virulence decrease was primarily attributed to loss of appressorium formation. However, reduction in ROS-generating enzymes expression such as NOX complex, IQG1, PLS1 and SOD1 was also observed in *nop53*, suggesting this may have contributed to further virulence loss. Interestingly, *nop53* growth on cellulose-rich medium was impaired, indicating that NOP53 may control cellulase function.

Cell wall synthesis is essential for fungal growth and reproduction. A minor component of *B. cinerea* cell wall is the sugar residue rhamnose (RHA) [74]. To examine the contribution of RHA in *B. cinerea* biology, the RHA biosynthetic genes glucose dehydratase *DH* and deoxyglucose epimerase reductase *ER* were disrupted. As expected, single *dh* and *er* mutants and double mutant *dh er* had no RHA production. However, *dh* single and *dh er* double mutant displayed WT-like phenotype while *er* exhibited varying development and virulence defects. *er* suffered from decreased hyphal growth, conidiation, sclerotia formation, virulence and increased cell wall stress sensitivity. It was suggested that the production of intermediate UDP-4-keto-6-deoxy-glucose (UDP-KDG) may have accumulated and been integrated into the fungal cell wall, causing defects in *er*.

The Plasma Membrane ATPase Related (*PMR1*) gene encodes a P-type ATPase used to transport calcium and manganese into the Golgi compartment for protein glycosylation. Deletion mutant *pmr1* exhibited defective growth, conidiation and lesion expansion in host tissues but increased sclerotia formation [94]. Analysis of cell wall composition revealed a significant reduction of glycoproteins with phosphomannan groups and increased chitin and glucan proportion. Interestingly, biofilm formation of *pmr1* was compromised, indicating PMR1 plays essential roles in host colonization. The alteration in cell wall composition might be the cause of impaired *pmr1* development. Another P-type ATPase in *B. cinerea* is the Copper-transporting ATPase CCC2, which is used to transport copper into the fungi for copper-containing proteins formation [83]. Deletion of *CCC2* resulted in mutants that formed fewer sclerotia and conidia and developed fewer appressoria with abnormal morphology. *ccc2* was completely avirulent and unable to cause lesions even in pre-wounded leaves, indicating that copper transporting ATPases are required for *B. cinerea* pathogenicity.

Kynurenine 3-monooxygenase (KMO) catalyzes the first step of the Kynurenine pathway, the primary synthesis pathway of the essential cofactor nicotinamide adenine dinucleotide (NAD). T-DNA insertion mutant of *KMO* resulted in reduced growth, failure to produce sclerotia or conidia but increased virulence [124]. An increase in CWDEs and phytotoxin secretion was observed in the mutant, suggesting that KMO may negatively regulated their activities. Surprisingly, expression of key genes in the cAMP and MAPK signaling pathways was also affected by *KMO* deletion, indicating that KMO is involved in these signaling pathways in an unknown manner.

Aquaporins (AQPs) are channel proteins that mediates the influx of water across membranes. In *B. cinerea*, AQP8 is essential for H_2_O_2_ uptake and the deletion mutant *aqp8* exhibited suppressed growth, conidiation and sclerotia formation [98]. Interestingly, *AQP8* deletion downregulated *NOXR* expression and the observed developmental defects may be caused by the low NOXR level. A loss of oxidative burst was also observed in *aqp8* mutant, suggesting AQP8 may be required to induce ROS-mediated pathways through indirect interaction with ROS generating members such as RHO3 and SOD1.

Autophagy-related genes (*ATG*) encode proteins essential for autophagy, a conserved process of degrading intracellular molecules and organelles by engulfing and lysosome fusion. Autophagosome assembly is initiated by serine/threonine kinase ATG1, whereby substrates later targeted by ubiquitin-like (UQL) autophagy marker ATG8 [60,93]. ATG8 is post-translationally modified by cysteine protease ATG4 [64] and utilities UQL activating enzymes ATG3 and ATG7 to target substrates [104]. Single deletion of aforementioned *ATG* genes resulted in no autophagy or sclerotial formation and significantly retarded conidiation, hyphal growth and virulence. Growth was only suspended in deletion mutants after a few days and it is suspected to be due to waste product accumulation. Interestingly, sclerotia formation defects were rescued in *atg3* and *atg7* mutants by exogenous nutrient addition, indicating the defect might be the result of nutrient deficiency [104]. This is further supported by *atg8* mutant grown in medium lacking nitrogen sources (MM-N) exhibiting delayed conidia germination [60]. No symptoms were observed on host tissues inoculated with any deletion mutants but necrosis appeared slightly in pre-wounded host tissues, indicating that loss of infection structure formation contributes to virulence reduction. The other causes of virulence loss remain unclear. Intriguingly, lipid droplet composition was significantly reduced in all mutant conidia, resulting in impaired conidiation. In sum, *ATG* genes play significant roles in *B. cinerea* development and pathogenicity.

The subtilisin-like serine proteases (SERs) have been shown to play significant development and virulence roles in pathogenic fungi. In a recent study, single and double mutants of *B. cinerea ser1*, *ser2* and *ser1 ser2* were generated [103]. While *ser1* mutant displayed a WT-like phenotype, *ser2* and *ser1 ser2* had impacted growth, conidiation, sclerotia formation and virulence. Although growth of *ser2* and *ser1 ser2* mutants was only slightly decreased, no conidia or sclerotia production was observed. *ser2* and *ser1 ser2* could only infect wounded host tissues for lack of the infection cushions (compound appressoria). Based on the genetic data, SER2 is essential in *B. cinerea* development and pathogenicity, particularly for conidiation and sclerotial formation [156]. Subcellular localization experiments may help in determining the presence of interactions between SER1, SER2 and ATG proteins.

The exocyst complex serves a role in post-Golgi vesicle trafficking and tethers to the plasma membrane. The exocyst complex component EXO70 is essential for vesicle docking and fusion in yeast [157]. Deletion of *EXO70* gene resulted in significantly decreased growth, condition and sclerotia formation in *B. cinerea* [79]. Microscopy of *exo70* mutants revealed enlarged and scattered vesicles, indicating EXO70 is essential for normal vesicle formation and transportation. The impaired growth and decreased CWDEs secretion may have contributed to the reduced virulence of *exo70*. Interestingly, exocytosis was still observed in *exo70*, suggesting the existence of alternative exocyst complex-independent exocytotic mechanisms.

F-actin capping protein (CP) regulates actin filaments by binding to the barbed ends (fast growing ends). In *B. cinerea*, the CP consists of an α and β subunits heterodimer. Deletion of CP α subunit gene (*CPA1*) resulted in impaired growth, penetration ability and conidia germination and few conidia and sclerotia [127]. Growth was further decreased in light conditions, indicating a stress resistant role of CPA1. Investigating the interactions between CPA1 and the CP β subunit may reveal further properties of CPA1.

The *STR2* gene encodes cystathionine γ-synthase, a key enzyme in methionine (Met) biosynthesis pathway responsible for converting cysteine into homocysteine [63]. *str2* deletion mutant exhibited severely impaired vegetative growth, pathogenicity, differentiation and stress resistance. The decreased growth rate stems from Met deficiency, as *str2* growth defect could be rescued by exogenous Met or homocysteine application. *str2* produced fewer conidia and no sclerotia, indicating a role of STR2 in fungal development. The mutant also exhibited increased sensitivity to oxidative, osmotic and cell wall damaging agents, suggesting interactions between Met synthesis and regulatory pathways such as AOX or MAPK pathways. Interestingly, the *str2* mutant could not even cause lesions in pre-wounded host tissues. It remains unknown what the virulence factors are affected to cause the complete loss of pathogenicity.

The heterodimer KU70/KU80 is the first key component of the nonhomologous end-joining (NHEJ) pathway, a mechanism of DNA repair [90]. *ku70* and *ku80* deletion mutants did not exhibit any development or virulence changes from WT strains. However, there were vastly improved genes inactivation rates in *ku70* and *ku80* deletion mutants, particularly for genes normally difficult to knockout in WT strains. Therefore, deleting *ku70* and *ku80* may assist in the success and efficiency in generating *B. cinerea* deletion mutants of other genes.

### 3.6. Melanization and Its Effects on Development and Pathogenicity

In fungi, production of dihydroxynaphthalene (DHN) melanin—a dark-colored pigment—is essential for their survival against abiotic stress such as ROS, desiccation, temperature fluctuations, UV rays and fungicides [158]. Consequently, there is strong interest in studying the biosynthesis and functions of melanin in *B. cinerea*. In particular, melanin is concentrated in sclerotia of *B. cinerea*, with smaller amounts found in the conidia and mycelia. We will first be discussing the melanin biosynthetic genes.

The DHN melanin biosynthetic pathway in *B. cinerea* (Figure 5) comprises of the polyketide synthase PKS12 and PKS13, tetrahydroxynaphthalene (TNH) reductase BRN1 and BRN2, scytalone dehydratase SCD1 and the hydrolase YGH1 [61]. During sclerotial melanization, the initial acetyl-CoA substrate is converted to 1,3,6,8-TNH (T4HN) by PKS12. T4HN is then catalyzed to scytalone by BRN2 and then dehydrated by SCD1 to 1,3,6-TNH (T3HN). T3HN is reduced and dehydrated by BRN1 and SCD1 sequentially to DHN, which is finally polymerized to DHN melanin by unknown enzymes. Interestingly, conidium and mycelium melanogenesis utilizes an alternative pathway of T4HN generation. In detail, acetyl-CoA is converted with PKS13 to acetyl-1,3,6,8-TNH (AT4HN) and then hydrolyzed by YGH1 to T4HN.

Deletion of the *PKS12* and *PKS13* genes resulted in albino sclerotia and conidia in *B. cinerea*, indicating complete melanogenesis disruption [61,80]. However, *pks13* mutants did not exhibit any changes in development or pathogenicity. The effects of PKS12 remains unclear due to conflicting reports on *pks12* mutant virulence, conidiation and hyphal growth phenotypes in the two individual studies. However, similarities between PKS13 and PKS12 structures support the idea that PKS12 is likely not involved in fungal development and virulence. As for TNH reductase BRN1 and BRN2, not only were *brn1* deletion mutants deficient in melanin biosynthesis but they also exhibited reduced conidiation and increased growth and virulence [80]. This was contradicted in another study, where sclerotia from *brn1 brn2* double mutants were similar to WT [61]. Although *brn1* mutant analysis has conflicting results, it is likely that the effects of *BRN1* deletion would be similar to other melanin biosynthesis genes and is nonessential for fungal development and pathogenicity. Deletion *scd1* mutant exhibited strong reduction in melanogenesis with no effects on other cellular processes [61]. Intriguingly, heterokaryotic *ygh1* deletion mutant exhibited decreased conidia formation [61], suggesting that either conidia melanization is essential for conidiation or PKS13 products (AT4HN) is detrimental to conidia formation. In fact, the latter seems more likely, as disruptions in later stages of melanin biosynthesis did not affect conidiation. In sum, studies so far indicated that melanization does not significantly affect fungal development and pathogenicity.

Studies have indicated that fungal melanin biosynthesis is controlled by MAPK pathways [159]. As mentioned in earlier sections, MAPK pathways regulate a vast number of gene expression. In *B. cinerea*, melanin biosynthesis is induced by the conserved CWI pathway, consisting of the cascade of BCK1 (Bypass of C Kinase), MKK1 and BMP3 (Botrytis MAPK 3) [52,108]. Single deletion mutants *bck1*, *mkk1* and *bmp3* all exhibited severely reduced pathogenicity, hyphal growth, conidiation and tolerance against cell wall and oxidative stress. *bmp3* mutant also failed to produce any sclerotia. Furthermore, melanization and expression of melanogenesis genes *SCD1* and *PKS13* were significantly downregulated, indicating that the CWI pathway induces melanin synthesis. Although sclerotia formation was not tested in *bck1* and *mkk1*, it is highly possible that these mutants also failed to produce sclerotia because they are upstream kinases of BMP3. Interestingly, virulence in *mkk1* was not decreased relative to *bck1* and *bmp3* because of increased OA secretion in *mkk1* [52]. Protein analysis revealed that MKK1 negatively regulates OA secretions by impeding RIM15 phosphorylation. The primary cause of defective pathogenicity appears to be from decreased appressium formation. PRO40 serves as a scaffold protein for the BCK1-MKK1-BMP3 cascade [52]. Disruption of the *PRO40* gene resulted in increased acid and hydrolase sensitivity and reduced activation of components in the CWI pathway, indicating PRO40 is essential for the function of the CWI pathway. Overall, the CWI pathway appears to be essential to *B. cinerea* development and pathogenicity and plays key roles in inducing melanogenesis gene expression. It remains unknown how the CWI pathway connects with other MAPK pathways.

In *B. cinerea*, protein tyrosine phosphatase PTPA and PTPB positively regulate phosphorylation of SAK1 and BMP3 [100]. Subsequently, single deletion of *PTPA* and *PTPB* genes resulted in reduced activation of the SAK1 and CWI pathway. Both deletion mutants exhibited defective sclerotium formation and virulence and had increased osmotic and oxidative stress sensitivity. Only *ptpA* had compromised conidiation. Interestingly, *ptpA* and *ptpB* melanin pigment was drastically increased despite suppression of the CWI pathway, indicating PTPA and PTPB are not upstream phosphatases of the CWI pathway. PTPA and PTPB may be negative regulators of melanin biosynthesis through unknown pathways. Similarly, deletion of Type 2C Serine/Threonine phosphatases (PP2Cs) *PTC1* and *PTC3* genes resulted in significantly impaired growth, sclerotium formation and virulence but increased conidiation [72]. Also, *ptc1* and *ptc3* deletion mutants have highly pigmented mycelia compared to WT. Furthermore, in vivo phosphorylation analysis in the study revealed that PTC3 is phosphorylated by SAK1, indicating PTC3 may be indirectly manipulating melanin biosynthesis through the MAPK-associated CWI pathway.

The regulators of DHN melanin biosynthetic gene expression appear to specifically control melanization of sclerotia and conidia separately. The sclerotial melanin regulator SMR1 regulates melanization of sclerotia [70]. Analysis of black colored (BS) and orange colored (OS) sclerotia mutants, named respectively after sclerotia appearance, revealed a mutation in *SMR1*, which resulted in downregulation of sclerotial melanogenesis genes including *PKS12, YGH1, BRN1/2, SCD1*. The loss of melanin in OS resulted in lower survival rates under all conditions and increased susceptibility to mycoparasites infection [160]. However, conidiation and conidial melaninization were not affected, suggesting that SMR1 is a positive regulator of genes only for sclerotial melanization. Change in development and pathogenicity in *smr1* was not tested.

Disruption of multiple broad regulators also severely affects melanization in *B. cinerea*. The deletion of the transcription elongator factor ELP4 resulted in a mutant with decreased growth, virulence and sensitivity to osmotic and oxidative stress [121]. No sclerotia were formed in *elp4*. Hyphal and conidia melanization were reduced, which was confirmed by expression analysis of downregulated *PKS13* and *SCD1*. The change in cell wall protein and polysaccharide composition may have contributed to the reduction in growth and virulence. Another mutation affecting conidial melanization was the *BAG1* gene deletion. BAG1 belongs to the diverse Bcl-2 associated athanogene (BAG) family, a group of co-chaperones which assists chaperones in their molecular functions [114]. *bag1* deletion mutants exhibited enhanced hyphal and conidial melanin accumulation, decreased pathogenicity and conidiation and failure to form sclerotia. Virulence loss stemmed from reduction in penetration structure formation and conidia germination. Increased sensitivity to salt, cell wall and temperature stress was also observed in *bag1*. Yeast-two hybrid assays revealed BAG1 negatively regulates the unfolded protein response (UPR) and the CWI pathway. Contrary to the assumption that melanization reduces stress sensitivity, *bag1* deletion mutants displayed increased environmental stress sensitivity. This may be because ELP4 and BAG1 both regulate the CWI pathway, which would have affected the integral composition of the fungal cell wall. The effects of melanin in *B. cinerea* remain unclear because melanization disruptions in most cases also interfere with the CWI pathway.

The VELVET complex plays various roles in the regulation of virulence, light-dependent development and melanization in fungi by acting as transcription factors for fungal development and SM gene expression. Currently, the VELVET proteins VEL1, VEL2 and VEL3 (VEL1 and VEL2 is also known as VEA and VELB respectively) have been studied in *B. cinerea* [48,49,131]. Deletion of *VEL1* and *VEL2* genes resulted in increased mycelial melanization and conidiation but reduced hyphal growth, sclerotial formation and lesion expansion. Studies on *vel1* and *vel2* secretome revealed that the mutants secreted significantly less citric acid and OA, which is suspected to be the primary cause of decreased pathogenicity [48,161]. The increased melanization is due to increased expression of melanin biosynthesis genes [49,131]. Deletion *vel3* mutant developed both conidiophores and sclerotia in darkness, indicating the involvement of the VELVET complex in light sensing processes [48]. A VELVET-like complex is also formed by VEL1, VEL2 and methyltransferase loss of AFLR expression (LAE1). Deleting the LAE1 gene resulted in mutants with reduced pathogenicity, light independent conidiation and sclerotial formation loss [161]. Similar to VELVET mutants, melanogenesis genes expression was increased in *lae1*. The cause of reduced virulence is likely due to reduced expression of SM synthesis enzymes and CWDE genes in *lae1*. In sum, the *B. cinerea* VELVET complex negatively regulates melanization and is essential for light-dependent development, acid secretion and virulence factor secretion. However, the role of melanization cannot be determined from VELVET complex mutants, as the mutant phenotypes are caused by disruption of multiple processes in *B. cinerea*.

## 4. Conclusions

Over the past decade, many genes involved in the development and pathogenicity of *B. cinerea* have been characterized. Accordingly, pathogenicity mechanisms and virulence factors were discovered, whereby *B. cinerea* suppresses plant defense to establish host infection and invasion. Degradation of plant tissues is achieved via the production of acids, ROS, CWDEs and compound appressoria and inducing plant cell death.

With almost 11,000 genes encoded in its genome, only less than 150 genes have been genetically studied so far through mutant analysis. This explains our limited understanding of this infamous fungus and the lack of knowledge about the pathways of its unique biological mechanisms. Besides the traditional targeted gene disruption methods, recent innovations in genetic analysis approaches such as CRISPR genome editing [162], should be utilized to enhance studies on this important pathogen.

Due to the interconnectivity of the signaling pathways in *B. cinerea*, it has been challenging to identify the specific functions of virulence factors with mutant analysis. Future improvements in genetic and biochemical tools for *B. cinerea* specific studies may aid researchers to unveil the mysteries of this fungus. Unfortunately, because of the complex pathogenicity mechanisms of *B. cinerea*, generating resistant cultivars may be sophisticated without causing unwanted side effects. Likewise, the high genomic diversity makes it difficult to develop effective long-term fungicide control methods. Further collaboration between molecular pathologists and resistance breeders will be crucial to understand the pathogenic mechanisms and host resistance behind this pathogen.

## Figures and Tables

**Figure 1 pathogens-09-00923-f001:**
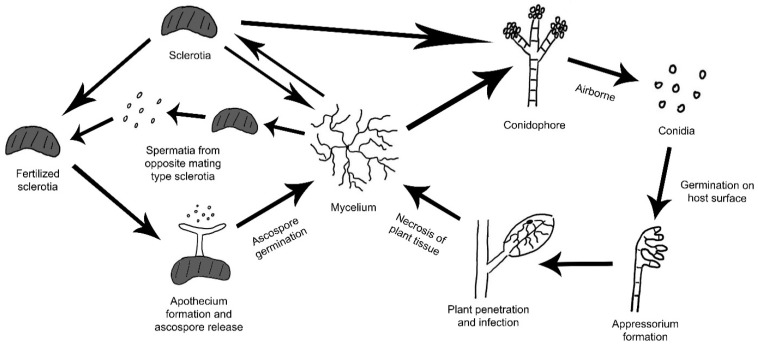
Life cycle of *Botrytis cinerea.*

**Figure 2 pathogens-09-00923-f002:**
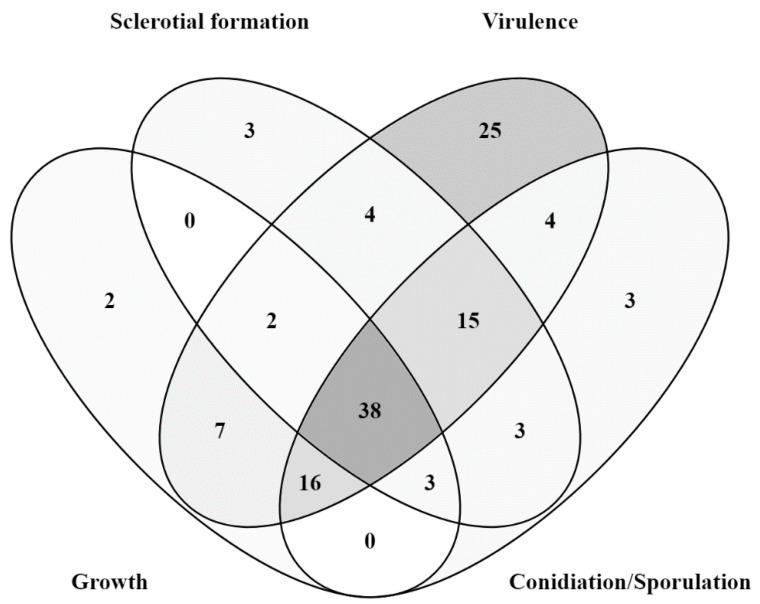
A Venn Diagram summary of *B. cinerea* genes studied with mutant analysis (detailed in Table 1).

**Figure 3 pathogens-09-00923-f003:**
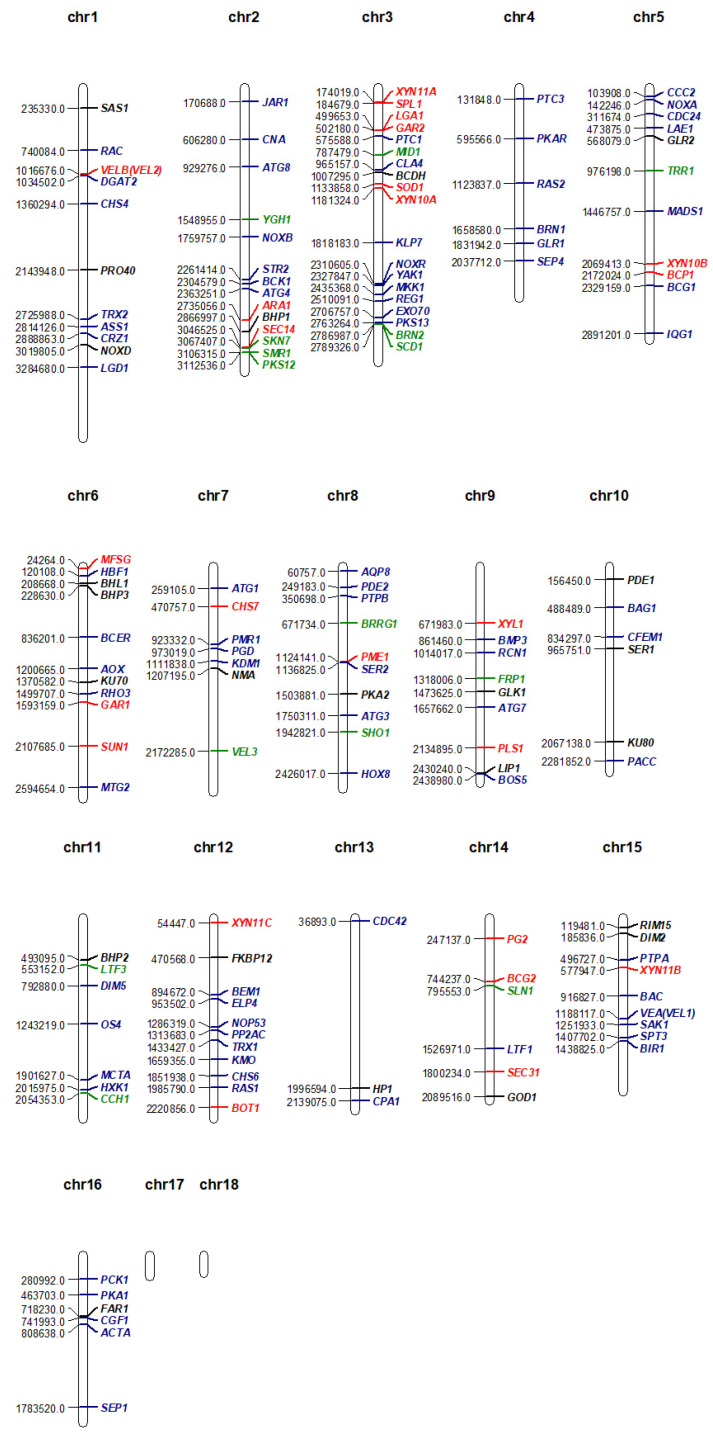
Map positions of the genes studied in the *B. cinerea* genome through mutant analysis. The numbers on the left of each chromosome represent the locations of these genes. Genes labelled in green are involved in development, including hyphal growth, sclerotial formation, conidiation and so forth. Genes labelled in red mainly play roles in virulence. Genes labelled in blue are involved in both while the ones labelled in black are involved in other biological processes. The chromosomal map was drawn using ‘MapChart’ software using information from Table 1.

**Figure 4 pathogens-09-00923-f004:**
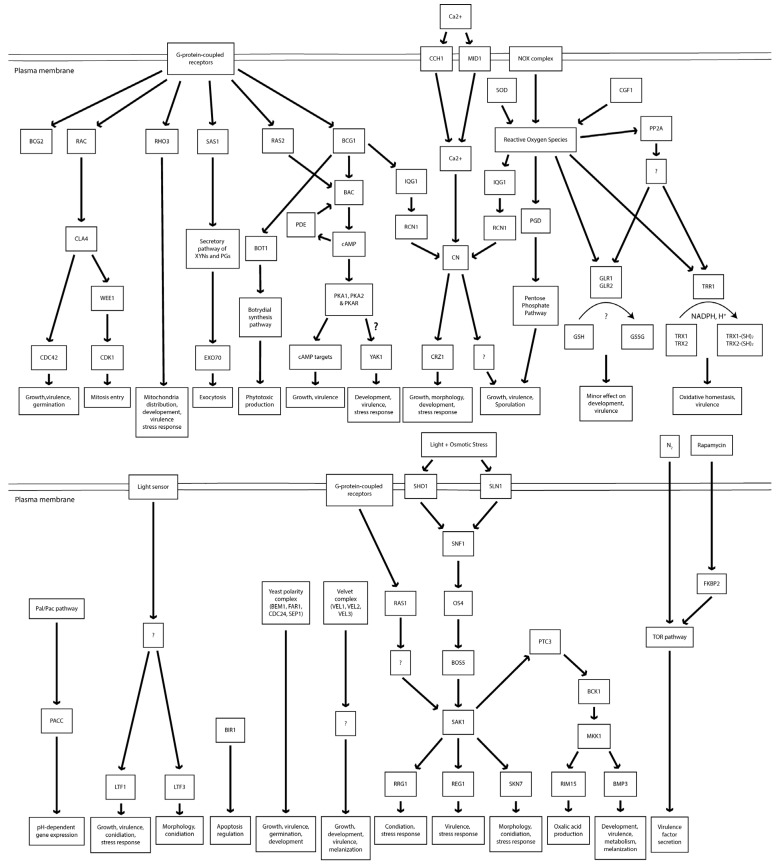
Signaling pathways in *B. cinerea* (detailed in Table 1).

**Figure 5 pathogens-09-00923-f005:**
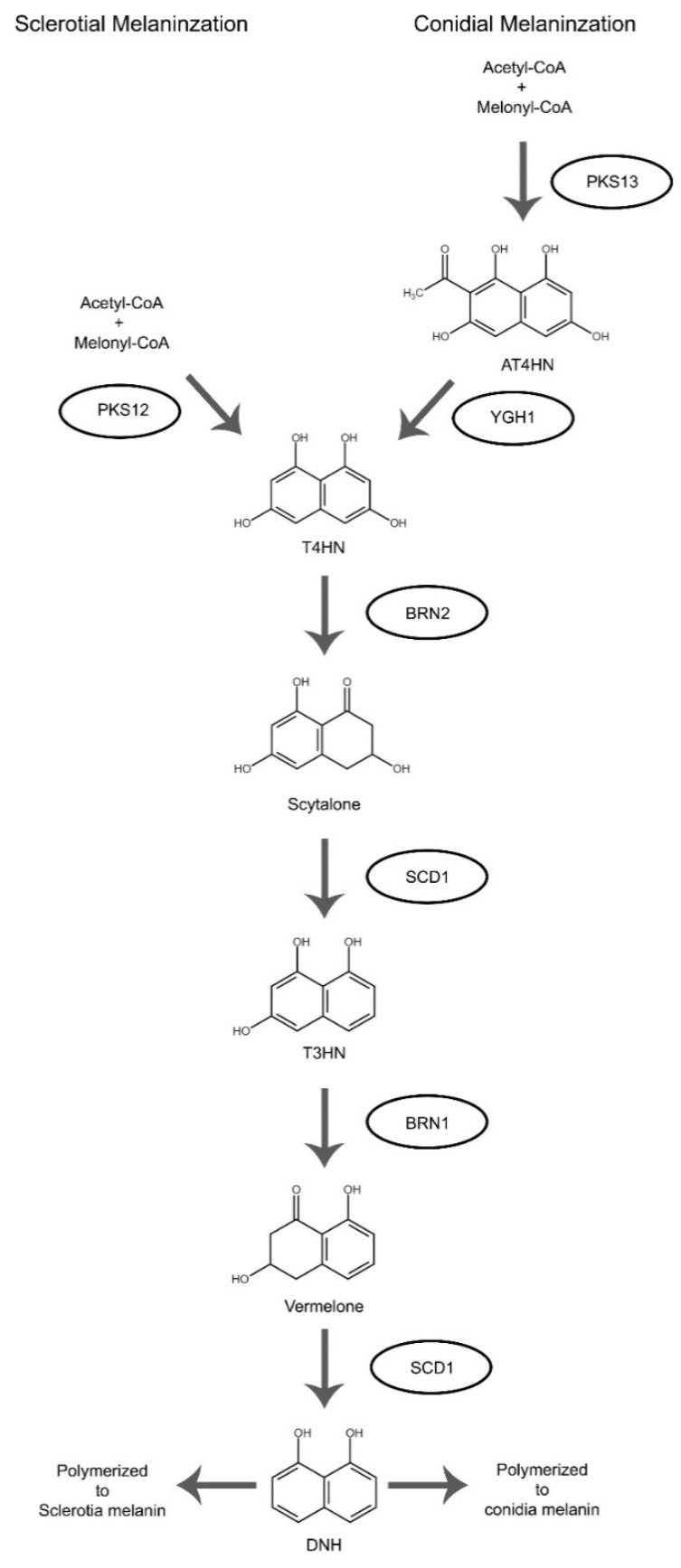
Dihydroxynaphthalene (DHN) melanin synthesis pathway in *B. cinerea.*

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
