# Peer review of "The Destructive Fungal Pathogen Botrytis cinerea—Insights from Genes Studied with Mutant Analysis"

_pathogens, 2020, doi:10.3390/pathogens9110923_

Round 1
Reviewer 1 Report
This review article indicates the knowledge about the biological features and the pathological mechanisms of Botrytis cinereal. This article is substantial. However, there are few illustrations which could support our understandings. The authors should have shown the illustrational abstracts regarding signaling mechanisms in the sections of 3-a, b, c, d, e.
Author Response
Reviewer #1:
This review article indicates the knowledge about the biological features and the pathological mechanisms of Botrytis cinereal. This article is substantial. However, there are few illustrations which could support our understandings. The authors should have shown the illustrational abstracts regarding signaling mechanisms in the sections of 3-a, b, c, d, e.
We made a new Figure 4 to summarize the diverse signaling events reviewed.
Reviewer 2 Report
Dear authors,
The manuscript “The destructive fungal pathogen Botrytis cinerea: insights from genes studied with mutant analysis” reviews the current knowledge about B. cinerea effectors and infection signalling events, through a fungus mutant analysis point of view. This manuscript is structured in 3 sections (and sub-sections) that include an introduction to the subject, B. cinerea reference genome and molecular dissection of B. cinerea biology. It also includes an extensive table that summarizes the genes from B. cinerea that have been studied using mutant analysis. This information could be a significant resource to future research not only in Botrytis but also in other necrotrophic pathogens.
Furthermore, this manuscript includes significant number of references from the last five years, many of them from 2019 and 2020, as well as relevant references related to the subject.
I recommend minor suggestions to the authors that I think will improve the manuscript
Best regards
Please consider the following suggestions:
Line 44 – replace “dried” by “mature”
Line 45 – replace “infected tissues and seeds” by “infected tissues as well as seeds”
Line 49 – please insert a reference at the end of the phrase
Line 60/61 – B. cinerea is a necrotrophic pathogen. Therefore, I suggest eliminating the phrase “After initial …. the plant”. Otherwise include a reference that supports our affirmation.
Line 89 – replace “post harvesting” by “post-harvesting”
Line 107 - Section “genomic sequences”: there are two Botrytis reference genomes available (strain B05.10 and T4), but only one is referred (B05.10) in this section. However, in table 1 and well as in several parts of the manuscript (such as in line 698), strain T4 is referred. Why the authors made this choice? If it was intentional, please clarify in the section the intention of only considering strain B05.10.
Line 185 – A venn diagram with the encoded proteins was provided, but no explanation was given. Could the author include a conclusion from this diagram (similar to what was done for chromosome map position)? Please restructure.
Line 209 – include a reference to support our affirmation “Over the last two decades, accessibility to the B. cinerea whole genome sequence and advancements in mutant analysis techniques have greatly improved molecular studies of this fungus”
Line 257 /258 – Something is missing in this phrase. Perhaps could be “However, the exact molecular mechanisms of PKAR and PKA1 regulation/interaction are not well established”
Line 323 – please include a reference to support the phrase “Protein kinases are critical to intracellular functions because of their ability to phosphorylate components of signal transduction systems to transduce signals”
Line 401/402 – “As conidiation requires large amounts of nutrients acquired from host tissue invasion, nutrient availability is inheritably linked with fungal growth”. Although this is true, it is not always true. Please include a reference to support this affirmation.
Line 887/888 – no need to start a new paragraph
Table 1 – In host species, please include the correct host name (common our Latin name) as is in the caption. Example, in the table host name is “benthi” and in table footnote (line 247) is Nicotiana benthamiana.
Reference number 31 – please display the reference according to the journal author instructions.
Author Response
Reviewer #2:
The manuscript “The destructive fungal pathogen Botrytis cinerea: insights from genes studied with mutant analysis” reviews the current knowledge about B. cinerea effectors and infection signalling events, through a fungus mutant analysis point of view. This manuscript is structured in 3 sections (and sub-sections) that include an introduction to the subject, B. cinerea reference genome and molecular dissection of B. cinerea biology. It also includes an extensive table that summarizes the genes from B. cinerea that have been studied using mutant analysis. This information could be a significant resource to future research not only in Botrytis but also in other necrotrophic pathogens.
Furthermore, this manuscript includes significant number of references from the last five years, many of them from 2019 and 2020, as well as relevant references related to the subject.
I recommend minor suggestions to the authors that I think will improve the manuscript
Please consider the following suggestions:
All the following suggested corrections are made in the new version, please see track changes. Missing references were added.
Line 44 – replace “dried” by “mature”
Corrected.
Line 45 – replace “infected tissues and seeds” by “infected tissues as well as seeds”
Corrected.
Line 49 – please insert a reference at the end of the phrase
Reference added.
Line 60/61 – B. cinerea is a necrotrophic pathogen. Therefore, I suggest eliminating the phrase “After initial …. the plant”. Otherwise include a reference that supports our affirmation.
Reference added.
Line 89 – replace “post harvesting” by “post-harvesting”
Corrected.
Line 107 - Section “genomic sequences”: there are two Botrytis reference genomes available (strain B05.10 and T4), but only one is referred (B05.10) in this section. However, in table 1 and well as in several parts of the manuscript (such as in line 698), strain T4 is referred. Why the authors made this choice? If it was intentional, please clarify in the section the intention of only considering strain B05.10.
We added explanations at line 116-119 for clarification.
Line 185 – A venn diagram with the encoded proteins was provided, but no explanation was given. Could the author include a conclusion from this diagram (similar to what was done for chromosome map position)? Please restructure.
Explanation is added at line 189.
Line 209 – include a reference to support our affirmation “Over the last two decades, accessibility to the B. cinerea whole genome sequence and advancements in mutant analysis techniques have greatly improved molecular studies of this fungus”
Reference added.
Line 257 /258 – Something is missing in this phrase. Perhaps could be “However, the exact molecular mechanisms of PKAR and PKA1 regulation/interaction are not well established”
Mistake corrected.
Line 323 – please include a reference to support the phrase “Protein kinases are critical to intracellular functions because of their ability to phosphorylate components of signal transduction systems to transduce signals”
Reference added.
Line 401/402 – “As conidiation requires large amounts of nutrients acquired from host tissue invasion, nutrient availability is inheritably linked with fungal growth”. Although this is true, it is not always true. Please include a reference to support this affirmation.
Reference added.
Line 887/888 – no need to start a new paragraph
Paragraphs merged as suggested.
Table 1 – In host species, please include the correct host name (common our Latin name) as is in the caption. Example, in the table host name is “benthi” and in table footnote (line 247) is Nicotiana benthamiana.
Mistakes corrected in the table.
Reference number 31 – please display the reference according to the journal author instructions.
Corrected.
Round 2
Reviewer 1 Report
The reviewer appreciates the response and effort which has been done by the authors to enhance the quality of the work.